# CODE DRIVEN PLANNING WITH DOMAIN-ADAPTIVE SELECTOR

**Zikang Tian**[1,2]**, Shaohui Peng**[3,*]**, Di Huang**[1]**, Jiaming Guo**[1]**, Ruizhi Chen**[3]**,**
**Rui Zhang**[1]**, Xishan Zhang**[1,4]**, Yuxuan Guo**[1,5]**, Zidong Du**[1]**, Qi Guo**[1]**,**
**Ling Li**[2,3]**, Yewen Pu**[6]**, Xing Hu**[1]**, Yunji Chen**[1,2,*]

[1]State Key Lab of Processors, Institute of Computing Technology, Chinese Academy of Sciences
[2]University of Chinese Academy of Sciences
[3]Intelligent Software Research Center, Institute of Software, Chinese Academy of Sciences
[4]Cambricon Technologies
[5]University of Science and Technology of China
[6]Nanyang Technological University

## ABSTRACT

Large Language Models (LLMs) have been widely adopted as task planners for AI agents in sequential decision-making problems, leveraging their extensive world knowledge. However, the gap between their general knowledge and environment-specific requirements often leads to inaccurate plans. To address this, existing approaches rely on frequent LLM queries to iteratively refine plans based on immediate environmental feedback, which incurs substantial query costs. However, this refinement is typically guided by short-term environmental feedback, limiting LLMs from developing plans aligned with long-term rewards. We propose **Co**de Driven **P**lanning w**i**th Domain-Adaptive Sele**C**tor (CoPiC). Instead of relying on frequent queries, CoPiC employs LLMs to generate a diverse set of high-level planning programs, which iteratively produce and refine candidate plans. A trained domain-adaptive selector then evaluates these candidates and selects the one most aligned with long-term rewards for execution. Using high-level planning programs as planner and domain-adaptive selector as estimator, CoPiC improves planning while significantly reducing query costs. Results in ALFWorld, NetHack, and StarCraft II Unit Building show that CoPiC outperforms advanced LLM-based baselines, achieving an average (1) 19.14% improvement in success rate and (2) 79.39% reduction in token costs.

## 1 INTRODUCTION

Pre-trained on web-scale data, Large Language Models (LLMs) have shown remarkable potential in zero-shot learning, commonsense understanding, and contextual reasoning (Bai et al., 2022; Touvron et al., 2023a;b). Consequently, recent studies have increasingly explored the use of LLM-based planners to complete tasks in various environments like household scenarios (Brohan et al., 2023; Liang et al., 2023) and games (Wang et al., 2023a;b). These planners decompose high-level task instructions into coherent natural-language plans for environment execution. Compared to traditional learning-based AI agents, LLM-based planners significantly improve both efficiency and applicability, thanks to the advanced capabilities of LLMs.

However, the gap between general knowledge and specific environments often causes LLMs to have hallucinations, resulting in plausible yet infeasible plans involving non-existent objects or unavailable actions. To address this issue and facilitate grounded planning, especially for complex tasks involving multiple objects and diverse preconditions, most existing methods (shown on the left side of Figure 1) allow LLMs to iteratively generate and refine plans using immediate environmental observation, thereby improving their feasibility (Yao et al., 2022; Shinn et al., 2024; Wang et al., 2023b; Sun et al., 2024). However, the frequent step-by-step observations lead to extremely high

---

*Corresponding authors. Contact: `tianzikang21s@ict.ac.cn`

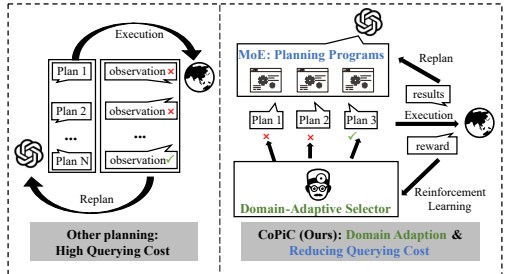

Figure 1: Difference between CoPiC and other planning paradigms: CoPiC leverages multiple high-level planning programs to reduce LLM's query costs, while a domain-adaptive selector is employed to select high-quality plans aligned with long-term rewards.

LLM's query costs. And the greedy search for plans by LLMs hampers the generation of long-term rewarding plans, reducing planning efficiency and further escalating query costs.

In summary, directly generating and refining static plans tailored to specific observations fails to adapt effectively to dynamic environments, resulting in high LLM's query costs. Inspired by the reliable code-generation capabilities of LLMs (Chen et al., 2021b; Hui et al., 2024; Guo et al., 2024), LLMs can potentially generate high-level planning programs that produces and refines plans based on varying observations, thus reducing query costs compared to static plans. However, the gap between LLMs' general knowledge and specific environments makes it challenging for a single planning program to handle all environmental observations. To address this, we introduce a mixture-of-experts (MoE) mechanism to generate multiple planning programs that produce diverse candidate plans, alongside a domain-adaptive selector that evaluates these candidates to select the plan most aligned with long-term rewards.

Inspired by these insight, we propose **Co**de-Driven **P**lanning w**i**th Domain-Adaptive Sele**C**tor (**CoPiC**, as shown on the right side of Figure 1). CoPiC comprises two core modules: an LLM planner and a domain-adaptive selector. The LLM planner generates multiple high-level planning programs that interact with environment, producing and refining candidate plans iteratively, which reduces the LLM's query costs incurring by generating and refining static plans step-by-step. The domain-adaptive selector evaluates these candidates and selects the one most aligned with long-term rewards, further bridging the gap between LLMs' general knowledge and environment-specific requirements. Upon termination of the planning programs, CoPiC leverages execution results to refine the planning programs and fine-tune the selector within a reinforcement learning framework, ensuring continuous improvement and adaptation to dynamic environments. This paper makes the following contributions:

- We propose a novel planning paradigm that combines an MoE mechanism, where each planning program acts as an expert, with a domain-adaptive selector to bridge the gap between LLMs' generality and environmental specificity.

- We propose CoPiC, a framework consisting of an LLM planner that generates multiple high-level planning programs and a domain-adaptive selector that selects the highest-quality plan, enhancing planning performance while reducing LLMs query cost.

- We evaluated CoPiC in three environments: ALFWorld (Shridhar et al., 2020), NetHack (Küttler et al., 2020), and StarCraft II Unit Building. The results demonstrate that CoPiC significantly outperforms advanced and immediate feedback-based baselines, including AdaPlanner (Sun et al., 2024), Reflexion (Shinn et al., 2024), Prospector (Kim et al., 2024), and REPL-plan (Liu et al., 2024b), achieving a 19.14% increase in success rate, a 79.39% reduction in query costs, and a 30.43% decrease in the average number of interaction steps. These findings highlight CoPiC's ability to deliver superior planning performance while reducing LLMs query overhead.

## 2 RELATED WORKS

CoPiC is LLM-based, code-driven and containing a selector scoring module. Therefore, we mainly review related works in these three areas. For a comprehensive overview, we also incorporate PDDL-based methods into the related works.

**LLM-based Planning** To bridge the gap between LLMs and the environment, prior works often adopt an immediate feedback paradigm, leveraging instant environmental feedback to refine plans. ReAct (Yao et al., 2022) and Inner Monologue (Huang et al., 2022) use step-by-step execution feedback to update actions, while Reflexion (Shinn et al., 2024) incorporates a trial-and-error mechanism, generating reflective texts from historical feedback. While these methods improve planning quality, they rely on extensive LLMs queries, resulting in high latency and limiting the generation of long-term rewarding plans. In contrast, CoPiC, which reduces costs through program-driven plan generation and improves domain adaptation via a selector module learned on domain experience.

**LLM-based Planning with Scoring** Some LLM-based planning methods use a Scoring Module to evaluate and select the best LLM-generated plans. For example, Prospector (Kim et al., 2024) uses direct LLM scoring or pre-trained models with offline expert data. SayCan (Brohan et al., 2023) assesses skill feasibility and integrates it with LLM planning, while SayCanPay (Hazra et al., 2024) further enhances efficiency by evaluating planning's payoff. Thus, current methods rely on two approaches: direct LLMs scoring (which lacks environmental priors and introduces scoring errors) or pre-training scoring models with offline expert data (which is costly and poorly generalizes to out-of-distribution data). In contrast, CoPiC introduces planning programs to dynamically and efficiently generate online data, enabling selector fine-tuning and scoring without these limitations.

**Program-Based Planning with LLMs** Extensive research on code generation using LLMs underpins the development of CoPiC. Key examples include Codex (Chen et al., 2021a) for Python function completion, AlphaCode for competitive programming, and studies like CodeT (Chen et al., 2022), Self-Debugging (Chen et al., 2023), and CodeRL (Le et al., 2022) that explore LLM debugging. Researchers have also applied LLMs to decision-making tasks, such as Code as Policies (Liang et al., 2023), PROGPROMPT (Singh et al., 2023), AdaPlanner (Sun et al., 2024), REPL-Plan(Liu et al., 2024b), and SDG (Peng et al., 2023). AdaPlanner(Sun et al., 2024) generates programmatic plans for each individual task, which are essentially static and lack dynamism. REPL-Plan (Liu et al., 2024b) leverages the Read-Eval-Print-Loop (REPL) tool, recursively invoking REPL through LLMs to generate new reusable APIs for planning. Unlike these methods, CoPiC produces multiple dynamic planning programs for each task category, creating a set of reasonable candidate plans. A domain-adaptive selector then samples the most rewarding plan, thus enhancing performance.

**PDDL-based Planning with LLMs** The Planning Domain Definition Language (PDDL) (Aeronautiques et al., 1998) is a human-readable, structured language for automated planning, defining possible world states, actions with prerequisites and effects, an initial state, and desired goals. It represents planning problems using a domain file (common elements across problems) and a problem file (specific initial states and goals). Recently, there has been growing interest in integrating traditional PDDL-based planning with LLMs, as seen in works such as (Silver et al., 2022), (Dagan et al., 2023), and (Silver et al., 2024). However, in these approaches, LLMs are primarily used for file completion or plan summarization, while actual planning relies on additional solvers to search the problem space for solutions. In contrast, CoPiC eliminates the need for external solvers by performing planning directly through programs generated by LLMs.

## 3 PRELIMINARY

### 3.1 PROBLEM FORMULATION

We explore the planning problem of LLMs-centric AI agent $\pi$ to address a variety of natural language described tasks $\mathcal{I}$ within specific environments. We first formulate the planning problem as a finite-horizon Partially Observable Markov Decision Process (POMDP) given by the tuple $\langle \mathcal{S}, \mathcal{O}, \mathcal{A}, R, P, \mathcal{I} \rangle$. Here, $\mathcal{S}$ is state space, $\mathcal{O}$ is a set of observations retrieved from states via an observation function $O : \mathcal{S} \to \mathcal{O}$, $\mathcal{A}$ is the set of actions, $R : \mathcal{O} \to \mathbb{R}$ is the reward function defined in environment, $P : \mathcal{S} \times \mathcal{A} \times \mathcal{S} \to [0, 1]$ is the stochastic transition function, $\mathcal{I}$ is the space of language described tasks.

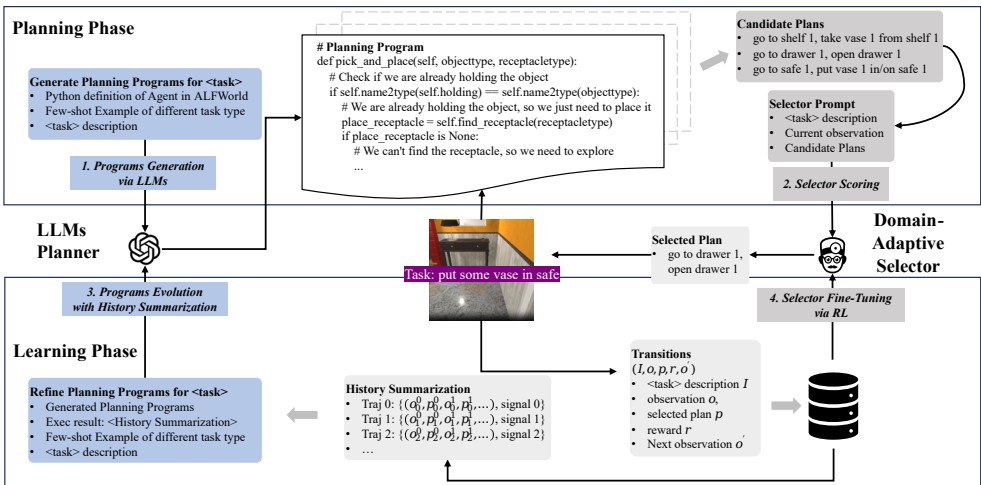

Figure 2: Overview of the CoPiC framework. CoPiC consists of two modules: an LLM planner that generates multiple planning programs to produce candidate plans and a domain-adaptive selector that selects the plan with the highest long-term reward. CoPiC alternates between the Planning Phase ("1. Programs Generation via LLMs" and "2. Selector Scoring") for generating and evaluating plans and the Learning Phase ("3. Programs Evolution with History Summarization" and "4. Selector Fine-Tuning via RL") for refining programs and improving the selector, progressively enhancing adaptability and performance.

Given an instruction $I \in \mathcal{I}$, the objective of the LLMs-centric AI agent $\pi$ is to find a plan $p$ to try to fulfill $I$:

$$\pi(p|I, o) : \mathcal{I} \times \mathcal{O} \rightarrow \Delta(\mathcal{A}^T) \tag{1}$$

where $o \in \mathcal{O}$ is the observation, $T$ signifies the total number of steps in the devised plan, $\Delta(\cdot)$ refers to the probability simplex function, and $p \in \Delta(\mathcal{A}^T)$ is an action sequence with length of $T$.

## 3.2 CoPiC

Previous mainstream paradigms (illustrated on the left side of Figure 1) address the aforementioned problem by enabling LLMs to generate, iteratively refine, and correct plans based on immediate environmental feedback. However, these approaches often incur substantial querying costs and limit the agent's ability to produce high-quality plans. In contrast, **CoPiC** employs efficient, low-cost planning programs generated by LLMs, coupled with a domain-adaptive selector, to produce and refine plans. This approach not only reduces querying costs but also improves the overall quality of the plans. Specifically, CoPiC initializes a set of planning programs $\{\rho_i\}_{i=1}^n$ utilizing LLMs. Each program $\rho_i$ is tasked with outputting a plan $p_i$ based on $I$ and $o$:

$$\{\rho_i(p_i|I, o) : \mathcal{I} \times \mathcal{O} \rightarrow \Delta(\mathcal{A}^T)\}_{i=1}^n \tag{2}$$

This results in a set of candidate plans $\{p_i\}_{i=1}^n$. To select the most domain-adaptive and holistic plan $p$ from these candidates, CoPiC introduces a domain-adaptive selector module:

$$C_\theta(p|o, I, \{p_i\}_{i=1}^n) \tag{3}$$

where $\theta$ denotes the parameters of the selector module. Consequently, the combination of the set of planning programs and the selector module forms our agent:

$$\pi_\theta(p|I, o) = C_\theta(p|o, I, \{\rho_i(p_i|I, o)\}_{i=1}^n) \tag{4}$$

By integrating the efficiency of planning programs with the effectiveness of the domain-adaptive selector, CoPiC significantly reduces LLMs querying costs while enhancing planning quality.

## 4 METHOD

As shown in Figure 2, CoPiC comprises two modules: an **LLM Planner** that generates multiple planning programs to produce candidate plans and a **Domain-Adaptive Selector** that evaluates and selects the plan with the highest long-term reward from candidates. CoPiC alternates between **Planning Phase** and **Learning Phase**.

During the Planning Phase, the LLM planner generates planning programs that iteratively interact with environment, dynamically adjusting and producing candidate plans at each step. The domain-adaptive selector then evaluates the candidates within the current context and selects the plan most aligned with long-term rewards. During the Learning Phase, execution results are used to refine the planning programs and fine-tune the selector within a reinforcement learning framework, enhancing domain adaptability. These two phases alternate to progressively optimize both planning programs and the selector. Detailed descriptions of these two phases are provided in the following subsections.

### 4.1 PLANNING PHASE

#### 4.1.1 GENERATING PROGRAMS TO PRODUCE PLANS

This stage, composed of planning programs generation and plans generation, utilizes LLMs to generate planning programs that produce and refine plans.

**Planning Programs Generation.** It begins by generating planning program $\rho(p|I, o)$ using "Init Prompt" to instruct the LLM, as shown in "1. Planner Generation via LLMs" Figure 2. The general structure of the prompt used for ALFWorld is presented in this figure, including the Python definition of ALFWorld, an example of a different task type, and the task description that needs to be completed. Considering a single planning program's limited sampling capability on plans may struggle with complex tasks, we generate $n$ $(n > 1)$ planning programs $\{\rho_i(p_i|I, o)\}_{i=1}^{n}$ for ensuring diverse plans. Detailed prompts for all environments can be found in Appendix H.

**Plans Generation.** Subsequently, each policy $\rho_i(p_i|I, o)$ takes the task instruction $I$ and the current observation $o$ of the environment to generate a plan $p_i$ (see planning program in Appendix I.3 for details on how plan is generated), forming a set of candidate plans $\{p_i\}_{i=1}^{n}$. **Note that the generated set of candidate plans has not yet interacted with the environment at this stage.**

To aid understanding, we have added an example of a planning program for 'Building SCV' in StarCraft II below. As we can see, the planning program is a program capable of interacting with the environment in a closed-loop paradigm. **The input to this program is the current observation from the environment, and the output plan is an action sequence generated based on that observation**. This plan is then used to interact with the environment to advance task completion.

```python
def planner(obs, action_space, task):
    '''
    obs is a dict with the specified number of each resource/building/unit at the current game state;
    action_space is a list of strings including all the available actions;
    task is a unit dict: {"SCV": num_1}, with the goal of building num_1 SCVs.
    '''
    plan_build, plan_unit, plan = [], [], []
    # infer the tech_tree for SCV
    tech_tree = {"SCV": {"base_building": "COMMANDCENTER", "pre_dependency": []}}
    # when supply_left is less than 8, increasing supply_cap (BUILD SUPPLYDEPOT) is necessary.
    if obs["Resource"]["supply_left"] < 8:
        if "BUILD SUPPLYDEPOT" in action_space: plan_build.append("BUILD SUPPLYDEPOT")
    # gas is important, check if there is a need to collecting gas (BUILD REFINERY).
    if obs["Resource"]["gas"] == 0 and "BUILD REFINERY" in action_space:
        plan_build.append("BUILD REFINERY")
    # Check the number of SCV that still need to be trained at the current game state.
    scv_still_needed_num = task["SCV"] - obs["Unit"]["SCV"]
    if scv_still_needed_num > 0 and f"TRAIN SCV" in action_space: plan_unit.append(f"TRAIN SCV")
    # Analyze which 'building' are still needed for buildinig SCV at the current game state.
    scv_base_building = tech_tree["SCV"]["base_building"]
    if obs["Building"][scv_base_building] < 1 and f"BUILD {scv_base_building}" in action_space:
        plan_build.append(f"BUILD {scv_base_building}")
    while plan_build or plan_unit:
        if plan_build: plan.append(plan_build.pop(0))
        if plan_unit: plan.append(plan_unit.pop(0))
    return plan
```

### 4.1.2 SELECTOR SCORING ON CANDIDATE PLANS

The Selector module then evaluates $\{p_i\}_{i=1}^{n}$ and samples the most adaptive and long-term rewarding plan $p$ to **interact** with the environment. The implementation of the Selector $C_\theta$ is inspired by TWOSOME(Tan et al., 2024). As illustrated in the "2. Selector Scoring" section of Figure 2, Selector $C_\theta$ is initialized from a tiny language model. Its scoring consists of three stages: calculation of plans' probability, plan text length regularization and normalization scoring of plan.

**Calculation of Plans' Probability.** The selector begins by taking the "Selector Prompt" as input, which includes the text description of the current observation, candidate plans, and a prompt instructing the selector to evaluate these plans. We denote the selector prompt as $d_{cp}$ and the text description of each plan from the planner as $d_{p_i}, i = 1, \ldots, n$. The plan description $d_{p_i}$ for plan $p_i$ can be represented as a sequence of tokens:

$$d_{p_i} = \{w_i^1, w_i^2, ..., w_i^{N_i}\}, i = 1, ..., n \tag{5}$$

where $N_i$ denotes the total number of tokens in $d_{p_i}$.

Subsequently, $C_\theta$ calculates the probability for the description of each plan in the context of the selector prompt, based on the probability of the corresponding tokens in that description:

$$\text{prob}(d_{p_i}|d_{cp}) = \prod_{k=1}^{N_i} \text{prob}(w_i^k|d_{cp}, w_i^1, ..., w_i^{k-1}), i = 1, ..., n \tag{6}$$

**Plan Text Length Regularization.** Due to the property of probability multiplication in language models, longer plans with more text inherently have lower likelihoods. Therefore, we apply the following regularization to eliminate the influence of plan text length:

$$\text{logit}(d_{p_i}|d_{cp}) = \log(\text{prob}(d_{p_i}|d_{cp})) \, / \, W_i, i = 1, ..., n \tag{7}$$

Here $W_i$ denotes the number of words in $p_i$. Based on the results from TWOSOME, we chose the number of words $W_i$ over the number of tokens $N_i$ for regularization.

**Normalization Scoring of Plan.** For language models, the sum of likelihoods for different sets of candidate plans is inconsistent; hence, $C_\theta$ scores plan $p_i$ by normalizing the logit of its description using a softmax function:

$$\text{score}(p_i) = \frac{\exp(\text{logit}(d_{p_i}|d_{cp}) \, / \, \text{Temperature})}{\sum_{j=1}^{n} \exp(\text{logit}(d_{p_j}|d_{cp}) \, / \, \text{Temperature})} \tag{8}$$

where Temperature is set to 1.0 for balancing exploration and exploitation during training and 0.0 for deterministic outputs during testing. Therefore, the score of each plan corresponds to the probability of its description. We then select the plan $p$ by sampling according to these probabilities:

$$p = C_\theta(I, o, \{p_i\}_{i=1}^{n}) \sim \{(p_i, \text{score}(p_i))\}_{i=1}^{n} \tag{9}$$

After executing $p$, the environment transitions to a new observation $o'$ and provides a reward $r$, forming a transition $(I, o, p, r, o')$ that is stored in a buffer, which is then used in the Learning phase.

## 4.2 LEARNING PHASE

### 4.2.1 PROGRAMS EVOLUTION WITH HISTORY SUMMARIZATION

The Programs Evolution with History Summarization iteratively improves the set of planning programs $\{\rho_i(p_i|I, o)\}_{i=1}^{n}$ by incorporating interaction history through in-context learning, as shown in the "3. Programs Evolution with History Summarization" section on the left side of Figure 2. It consists of history summarization and planning programs refinement.

**History Summarization.** After engaging with the environment across $N$ episodes, the history of the last $M$ episodes ($M \leq N$) is summarized in the format: $\{\langle \text{trajectory}_i, \text{signal}_i \rangle\}_{i=N-M+1}^{N}$. Here, $\text{trajectory}_i = (o_i^0, p_i^0, o_i^1, p_i^1, \ldots)$ represents the record of interactions, and $\text{signal}_i \in \{\text{True}, \text{False}\}$ indicates whether the task was completed in the $i$th episode.

**Planning Programs Refinement.** The successful examples, current planning programs, and history summarization are then integrated into a "Feedback Prompt", which enables the LLMs to evolve and generate an improved set of planning programs. The LLM is tasked with analyzing failed trajectories in comparison to successful ones (e.g., attempting to open a container without approaching it first). This comparative analysis reveals weaknesses in the planning programs, enabling targeted debugging (see the case analysis in Appendix F for a more intuitive understanding). By implementing this approach, the LLMs can progressively refine the set of planning programs.

### 4.2.2 SELECTOR FINE-TUNING VIA RL

Selector fine-tuning via reinforcement learning (RL) enhances domain experience utilization and ensures that the selector selects plans optimized for long-term rewards. To achieve this, CoPiC employs a parameter-efficient LoRA training architecture within the Proximal Policy Optimization (PPO) framework (Schulman et al., 2017) to fine-tune the Selector module.

**LoRA.** This process incorporates Low-Rank Adaptation (LoRA) (Hu et al., 2021) parameters and Multilayer Perceptron (MLP) layers to the final transformer block of the Selector's tiny language model. These components function as the actor and critic in the PPO setup, respectively.

**Fine-Tuning using PPO.** Transitions from the replay buffer are used to fine-tune the Selector according to the PPO objective (please see Eq 11 in Appendix). During fine-tuning, only the LoRA parameters and added MLP layers are updated, while the parameters of the language model itself remain frozen. The planning and learning phases is detailed in Algorithm 1.

## 5 RESULTS

| Method | Pick | | Examine | | Clean | | Heat | | Cool | | Pick Two | |
|---|---|---|---|---|---|---|---|---|---|---|---|---|
| | SR ↑ | Cost ↓ | SR ↑ | Cost ↓ | SR ↑ | Cost ↓ | SR ↑ | Cost ↓ | SR ↑ | Cost ↓ | SR ↑ | Cost ↓ |
| CoPiC (Ours) | **100.00** | **0.05M** | **100.00** | **0.04M** | **100.00** | **0.33M** | **100.00** | **0.26M** | **100.00** | **0.28M** | 95.29 | **0.06M** |
| CoPiC (TSL) | 100.00 | 0.04M | 100.00 | 0.04M | 100.00 | 0.29M | 96.52 | 0.27M | 100.00 | 0.25M | 100.00 | 0.09M |
| CoPiC (LaJ) | 84.17 | 0.08M | 78.89 | 0.09M | 78.71 | 0.42M | 80.87 | 0.33M | 78.10 | 0.37M | 75.29 | 0.11M |
| AdaPlanner | **100.00** | 0.63M | 64.44 | 1.95M | 91.61 | 0.89M | 76.52 | 0.74M | 89.52 | 1.57M | 87.06 | 1.58M |
| Reflexion | 91.67 | 2.09M | 86.67 | 1.51M | 73.55 | 2.54M | 75.65 | 1.88M | 73.33 | 1.46M | 81.18 | 1.65M |
| Prospector | 70.83 | 1.74M | 65.56 | 1.18M | 75.48 | 2.15M | 82.61 | 1.48M | 81.90 | 1.18M | 71.76 | 1.30M |
| REPL-plan | 82.50 | 0.66M | 83.33 | 0.41M | 89.03 | 0.75M | 91.30 | 0.61M | 88.57 | 0.47M | **100.00** | 0.43M |
| LLM-DP | 91.67 | 0.61M | 95.56 | 0.21M | 92.90 | 0.78M | 92.17 | 0.81M | 90.48 | 1.17M | 81.18 | 1.08M |
| GPT-3.5 | 2.50 | 0.56M | 16.67 | 0.41M | 3.23 | 0.73M | 4.35 | 0.54M | 4.76 | 0.47M | 4.71 | 0.40M |
| Cot-Zero-Shot | 4.17 | 0.72M | 5.56 | 0.55M | 4.52 | 0.96M | 7.83 | 0.71M | 4.76 | 0.65M | 10.59 | 0.52M |
| Cot-Few-Shot | 16.67 | 0.92M | 11.11 | 0.68M | 6.45 | 1.24M | 17.39 | 0.85M | 14.29 | 0.78M | 11.76 | 0.65M |
| BUTLER | 50.00 | – | 39.00 | – | 74.00 | – | 83.00 | – | 91.00 | – | 65.00 | – |
| TWOSOME | 71.67 | 0.32M | 74.44 | 0.19M | 76.78 | 1.24M | 76.52 | 1.08M | 80.95 | 1.12M | 81.18 | 0.72M |

Table 1: Comparison of CoPiC and baselines in ALFWorld. M represents millions. CoPiC (TSL) denotes CoPiC (Test Set Learning). CoPiC (LaJ) denotes CoPiC (LLM-as-a-Judge).

We conducted extensive experiments across three environments: ALFWorld (Shridhar et al., 2020), NetHack (Küttler et al., 2020), and StarCraft II Unit Building. The results highlight that CoPiC: **1)** reduces querying costs while improving success rate (Sec 5.2), **2)** exhibits superior data efficiency (Sec 5.3), and **3)** supports open-source LLMs (Sec 5.4.1), underscores the significance of programs evolution (Sec 5.4.2) and the selector module (Sec 5.4.3).

### 5.1 EXPERIMENT SETUP

**Environments** 1) **ALFWorld**: A widely used benchmark in planning researches (Shinn et al., 2024; Sun et al., 2024; Kim et al., 2024; Liu et al., 2024b), comprising six complex household task types. 2) **NetHack**: A roguelike game renowned for its intricate mechanics, with representative tasks (Drink Water, Open Box/Chest, and Upgrade Exp Level to 3) designed to test agent's planning abilities. 3) **StarCraft II Unit Building**: A challenging resource management problem studied in prior work (Churchill & Buro, 2011; Tang et al., 2018; Elnabarawy et al., 2020; Vinyals et al., 2019), consisting of tasks with varying complexity: Easy (SCV and BattleCruiser), Medium (SCV, Thor, Banshee, and Raven), and Hard (SCV, SiegeTank, Medivac, VikingFighter, and Ghost). For a detailed introduction to the environment, including task specifics and reward function design, please

refer to Appendix C. **Note** that the reward function design is straightforward and does not incorporate any expert knowledge.

**Baselines** We selected baselines: (1) Immediate feedback-based LLM planning methods, including vanilla LLM planning (GPT-3.5 / GPT-4o), CoT-Zero-Shot, CoT-Few-Shot (Wei et al., 2022), **Reflexion** (Shinn et al., 2024), **AdaPlanner** (Sun et al., 2024), **Prospector** (Kim et al., 2024), **REPL-plan** (Liu et al., 2024b), and PDDL-based method LLM-DP(Dagan et al., 2023); See Sec 2 for difference of CoPiC with these baselines. (2) Non-LLM training-based methods, including PPO (Schulman et al., 2017) (see Appendix E.6 for PPO's results and its detailed analysis), BUT-LER (Shridhar et al., 2020) and RL-based LLM Policy TWOSOME (Tan et al., 2024).

**Metric** We evaluated CoPiC against the baselines in two aspects: (1) **Planning Quality**–Success Rate (**SR** ↑) and **Step** ↓. SR is the percentage of tasks successfully completed during testing (higher is better). Step is the number of environment interaction steps needed to complete tasks during testing (lower is better). Since both SR and Step reflect planning quality, we place the results related to Step in Appendix E.3. (2) **Planning Efficiency**–Token Cost (**Cost** ↓). The total token costs of language model queries required to complete tasks, reflecting method efficiency and cost (lower is better). **Note** that for CoPiC, the Cost includes the token costs of both LLMs and Selector's tiny language model.

**Setting** **1)** The number of planning programs is set to $n = 3$ in CoPiC. **2)** We use Tinyllama (Zhang et al., 2024) as the Selector's language model for its lightweight nature and impressive performance. **3)** For CoPiC and the baselines, GPT-3.5 serves as the base LLM for ALFWorld and StarCraft II Unit Building tasks. Given the complexity of NetHack (Jeurissen et al., 2024), GPT-4o is used as the base LLM. **4)** The results are mean values of 5 seeds, and standard deviation can be found in Table 10 in Appendix. Additional details on settings can be found in Appendix D.

## 5.2 CoPiC: 19.14% Higher SR, 79.39% Lower Cost

| Environment | Nethack | | | | | | StarCraft II Unit Building | |
|---|---|---|---|---|---|---|---|---|
| Method | Drink Water | | Open Box/Chest | | Upgrade Exp Level to 3 | | Hard | |
| | SR ↑ | Cost ↓ | SR ↑ | Cost ↓ | SR ↑ | Cost ↓ | SR ↑ | Cost ↓ |
| CoPiC (Ours) | **70.00** | **0.53M** | **65.00** | **0.42M** | **78.67** | **0.15M** | **100.00** | **0.06M** |
| AdaPlanner | 60.00 | 0.64M | 52.67 | 1.21M | 67.67 | 1.25M | 71.00 | 0.36M |
| Reflexion | 58.67 | 1.07M | 50.00 | 1.91M | 69.67 | 2.12M | 24.00 | 0.78M |
| Prospector | 55.33 | 1.01M | 48.00 | 1.78M | 65.00 | 1.98M | 68.00 | 0.68M |
| REPL-Plan | 54.00 | 0.74M | 46.00 | 1.20M | 63.00 | 1.11M | 58.00 | 0.39M |
| GPT-4o | 33.33 | 1.17M | 10.00 | 1.08M | 0.00 | 1.82M | 0.00 | 0.6M |
| Cot-Zero-Shot | 56.67 | 1.24M | 20.00 | 1.11M | 3.33 | 1.68M | 0.00 | 0.63M |
| Cot-Few-Shot | 33.33 | 1.38M | 16.67 | 1.52M | 10.00 | 2.47M | 17.00 | 0.69M |

Table 2: Comparison of CoPiC and baselines in Nethack and StarCraft II Unit Building.

As shown in Table 1 and Table 2, CoPiC outperforms the advanced baselines (Reflexion, AdaPlanner, Prospector and REPL-Plan) across three environments. **On average, CoPiC achieves a 19.14% increase in success rate (SR ↑), and a 79.39% reduction in token costs (Cost ↓).**

In ALFWorld, CoPiC achieved an **16.96% improvement in SR**, and an **83.76% reduction in Cost**. And compared with AdaPlanner, which depends on task-specific prompts, CoPiC leverages a domain-adaptive selector to holistically refine programs, eliminating the need for such customization. Additionally, we constructed **CoPiC(TSL)—that is, CoPiC with Test Set Learning**—in Table 1 to demonstrate that CoPiC can work under the same experimental setup as the baselines, namely Test Set Online Learning. Despite this, CoPiC(TSL) still achieved a success rate that was 17.17% higher than the baselines, while reducing token consumption by 87.16%.

For NetHack, CoPiC achieved an **13.72% improvement in SR**, and a **70.96% reduction in Cost**. For challenging tasks like "Upgrade Exp Level to 3", CoPiC reduced LLMs queries by 96.11% while improving SR by 10%, thanks to its ability to prioritize long-term rewarding strategies. Specifically, the Selector module enabled CoPiC to identify that preserving a pet's life is a more effective long-term strategy for defeating monsters, highlighting its domain adaptability. In contrast, even though the baselines explicitly include prompts to protect the pet, they still attack it for short-term rewards.

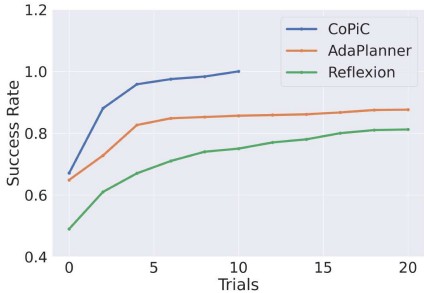

Figure 3: Comparison of SR varying with Trials for CoPiC, AdaPlanner and Reflexion on ALF-World.

| Open-Source LLMs | Method | SR | Cost |
|---|---|---|---|
| DeepSeek-Coder | CoPiC (ours) | **100.00** | **0.82M** |
|  | Adaplanner | 79.78 | 7.14M |
|  | Reflexion | 81.00 | 5.29M |
| DeepSeek-V3 | CoPiC (ours) | **100.00** | **0.89M** |
|  | Adaplanner | 76.81 | 7.87M |
|  | Reflexion | 83.00 | 7.94M |
| Qwen2.5-Coder-14B-Instruct | CoPiC (ours) | **99.76** | **0.80M** |
|  | Adaplanner | 64.29 | 1.82M |
| Qwen2.5-14B-Instruct | Reflexion | OOM | OOM |

Table 3: Average performance comparison using diverse open-source LLMs on ALFWorld. "OOM" indicates out-of-memory error.

In StarCraft II Unit Building, CoPiC demonstrated substantial improvements on Hard tasks, with a **44.75% increase in SR**, and an **87.86% reduction in Cost**. Results for Easy and Medium tasks further support its superior performance (see Appendix E).

In summary, CoPiC consistently outperforms all baselines across key metrics in diverse environments, demonstrating its ability to generate long-term rewarding plans while maintaining high efficiency. Unlike baselines like Reflexion, AdaPlanner and REPL-Plan, which rely on test-set learning, **CoPiC, trained solely on training tasks, generalizes effectively to unseen test tasks without incurring additional LLMs querying costs or requiring selector fine-tuning**. This unique zero-shot adaptation capability is enabled by its integration of planning programs and a domain-adaptive selector. Additionally, examples of evolved planning programs can be found in Appendix I and F.

### 5.3 CoPiC Requires Less Environmental Data

Figure 3 presents the average learning curves of CoPiC, AdaPlanner, and Reflexion on ALFWorld. **Note** that 'Trials' label on the x-axis is proportional to ($\propto$) the total number of interactions with environment, with a higher number of trials indicating more data. CoPiC demonstrates higher asymptotic performance while requiring less environmental data. The result indicates that, comparing to the immediate feedback mechanisms used in AdaPlanner and Reflexion, **CoPiC is more efficient and domain-adaptive, enabling the selection of plans with long-term rewards and ultimately delivering superior performance.**

### 5.4 Ablation

We conducted ablation studies to demonstrate that: **1)** CoPiC also supports open-source LLMs. **2)** The Programs Evolution module refines planning programs iteratively, enhancing overall performance. **3)** The Selector selects high-quality plans, thereby improving performance. Additional ablation studies on the impact of the number of planning programs are provided in Appendix E.4.

### 5.4.1 CoPiC Supports Open-Source LLMs

CoPiC supports both closed-source LLMs (e.g., GPT series) and open-source LLMs for generating and refining planning programs. We evaluated CoPiC, AdaPlanner, and Reflexion on the ALF-World benchmark using open-source LLMs like DeepSeek (Liu et al., 2024a; Guo et al., 2024) and Qwen2.5-14B (Yang et al., 2024; Hui et al., 2024). To ensure fairness, Qwen2.5-Coder-14B-Instruct was used for CoPiC and AdaPlanner, as both employ code-based methods, while Qwen2.5-14B-Instruct was applied to Reflexion, which follows a chat-style paradigm. As shown in Table 3, **CoPiC outperformed AdaPlanner by 26.29% in success rate while reducing cost with 85.09%**. Reflexion struggled with Qwen2.5-14B due to limited context handling, resulting in invalid responses, chat history accumulation, and eventual out-of-memory (OOM) errors. These results demonstrate CoPiC's superior compatibility and efficiency.

### 5.4.2 PROGRAMS EVOLUTION REFINES PLANNING

To evaluate the impact of programs evolution, we conducted an experiment where CoPiC's learned selector was frozen, existing planning programs were discarded, and the LLMs was tasked with generating and refining new planning programs using the frozen selector. The evolution curves are shown in Figure 4(a). As illustrated, **the average success rate (red curve) across three tasks (Clean, Heat, Cool) in ALFWorld improved from 75.60% (initial) to 91.44% (2nd iteration) and reached 100.00% (4th iteration).** These results underscore the pivotal role of programs evolution in refining planning programs and achieving high performance. Besides, a case of program evolution was provided in Appendix F.

### 5.4.3 SELECTOR SCORING ENHANCES PLAN QUALITY

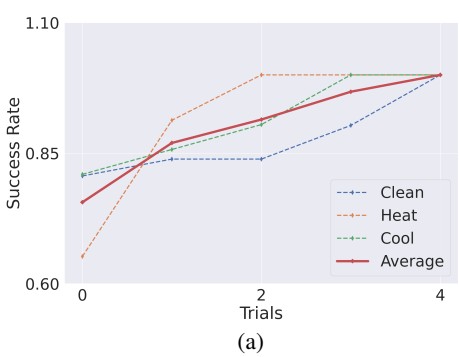
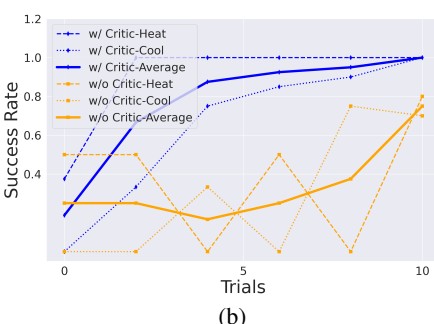

(a)                                                (b)

Figure 4: (a) Ablation on Programs Evolution: Evolution curves of program refinement for three ALFWorld tasks—Clean, Heat, and Cool—along with their average performance. (b) Ablation on Selector: Performance comparison of CoPiC with (w/) and without (w/o) the Selector in ALFWorld.

We implemented a variant of CoPiC without the selector, employing a strategy that randomly selects plans from the candidate plans generated by the planning programs. As shown in Figure 4(b), under the same cost (i.e., an equal number of interaction trials), **the success rate with a selector is consistently 20% to 60% higher than without a selector across two tasks (Heat and Cool) on average.** This demonstrates that the selector not only reduces LLMs querying costs but also enables the selection of higher-quality plans. In summary, these findings highlight the essential role of the selector in empowering the LLMs to generate high-performance planning programs. Additionally, in Table 1, we also constructed **CoPiC (LaJ), namely CoPiC with LLM-as-a-Judge**, to illustrate that the domain-adaptive selector presented in this paper outperforms an LLM lacking domain knowledge (here, GPT-4.1). Compared with LLM-as-a-Judge (**GPT-4.1-as-a-Judge**), the RL-finetuned selector improves the success rate by 19.88% and reduces token consumption by 27.14%. This improvement arises because the RL-finetuned selector acquires domain-specific knowledge during interaction, yielding more accurate plan evaluations than the prior-free LLM-as-a-Judge approach.

## 6 CONCLUSION AND DISCUSSION

In this paper, we propose **Co**de Driven **P**lanning w**i**th Domain-Adaptive Sele**C**tor (CoPiC), a novel planning framework that utilizes LLMs for complex tasks. CoPiC uses LLMs to generate multiple planning programs to iteratively refine plans, reducing the high query costs associated with step-by-step static plan refinement. And a domain-adaptive selector is adopted to evaluate these plans, selecting those best aligned with long-term rewards, further bridging the gap between LLMs' generality and environment-specific needs. We assess CoPiC across three challenging environments: ALFWorld, Nethack, and StarCraft II Unit Building. Our results show that CoPiC outperforms advanced baselines – Reflexion, AdaPlanner, Prospector and REPL-Plan – at a significantly reduced cost. Looking ahead, we are committed to expanding CoPiC's application to more complex tasks, including full games of StarCraft II and Civilization, as well as more intricate real-world scenarios. Additionally, we discussed the current limitations of CoPiC in Appendix B.

## 7 ETHICS STATEMENT

This work strictly adheres to the ICLR Code of Ethics. Our study did not involve any human subjects or animal experimentation. All benchmarks utilized, including ALFWorld, Nethack, StarCraft II Unit Building, and Robosuite, were sourced in full compliance with the relevant usage guidelines, ensuring no violations of privacy. We have taken meticulous care to avoid any biases or discriminatory outcomes throughout our research process. No personally identifiable information was used, and no experiments were conducted that could raise privacy or security concerns. We are committed to maintaining transparency and integrity throughout the entire research process.

## 8 REPRODUCIBILITY STATEMENT

We have taken extensive measures to ensure that the results presented in this paper are fully reproducible. The code for CoPiC, along with the planning programs we generated, has been included in the supplementary material. The experimental setup, including detailed configurations of hyperparameters, training and testing procedures, is meticulously described in Appendix D and Algorithm 1. We are confident that these resources and detailed documentation will enable other researchers to reproduce our work and build upon our findings to further advance the field.

## 9 ACKNOWLEDGEMENTS

This work is partially supported by the NSF of China (Grants No.62341411, 62525203, U22A2028, 62222214, 6240073476), Science and Technology Major Special Program of Jiangsu (Grant No. BG2024028), Strategic Priority Research Program of the Chinese Academy of Sciences (Grants No.XDB0660200, XDB0660201, XDB0660202), CAS Project for Young Scientists in Basic Research (YSBR-029) and Youth Innovation Promotion Association CAS.

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

## A    THE USE OF LARGE LANGUAGE MODELS (LLMs)

(1) We take LLM as the base model of CoPiC to generate planning programs. (2) We use LLMs to assist in checking and correcting grammatical and spelling errors in this paper.

## B    LIMITATIONS

To facilitate the evolution of planning programs and the training of the selector, CoPiC requires LLM to generate an initial planning program that can interact with the environment and acquire experience, even if it is not optimal. Consequently, the LLM used should possess adequate code generation capabilities. Additionally, since CoPiC is designed for planning-level tasks, scaling it to more complex and real-world scenarios – such as perception-planning-control tasks involving robots – requires integration with perception models and low-level controllers. We validated this scalability of CoPiC in Appendix E.7.

## C    ENVIRONMENTS DETAILS

### C.1    NETHACK

As shown in Netplay Jeurissen et al. (2024), the primitive Nethack with the goal to retrieve the Amulet of Yendor is impossible for LLM-based agent. Therefore, here we customized 3 potentially completable tasks based on NetHack: Drink Water, Open Box/Chest, and Upgrade Exp Level, thereby reasonably quantifying the performance of different approaches. **Drink Water** requires the agent to find a sink or fountain in the environment and drink from it. **Open Box/Chest** requires the agent to locate a box or chest in the environment and attempt to open it (e.g., using a key or by kicking). **Upgrade Exp Level to 3** requires the agent to kill monsters to reach level 3. All three tasks are still conducted in the original NetHack environment, where the agent must still pay attention to various states such as HP, hunger, and whether poisoned, etc., while also needing to defeat monsters to obtain food, equipment, and experience, etc. Therefore, these three tasks remain very challenging. All tasks use sparse reward functions, granting a reward of 1 only upon successful task completion and 0 otherwise.

### C.2    STARCRAFT II UNIT BUILDING

StarCraft II is a famous real-time strategy (RTS) game, which encompasses resource management, technological research, building order, and large-scale battles, all of which require strategic planning and quick decision-making. The game presents a multifaceted planning environment due to its high-dimensional action space, long-term planning horizon, and the need for both macro-management and micro-operation, thus offering a demanding yet fertile ground for AI advancement.

Among the many challenges in StarCraft II, building order is one of the pivotal ones, focusing on the types and orders of the buildings and units produced. An adeptly devised building order can markedly elevate the probability of triumph. How to construct an optimal building order is a sophisticated strategic dilemma, which has been explored through various methodologies, including heuristic search Churchill & Buro (2011), reinforcement learning Tang et al. (2018); Elnabarawy et al. (2020), and imitation learning Vinyals et al. (2019).

Therefore, taking into account the complexity nature of building order tasks, we designed a building benchmark based on StarCraft II. Given instructions describing a target unit collection, the agent needs to carefully plan resource collection, building sequence, and unit production until the task is completed.

#### C.2.1    DESIGN DETAILS

Specifically, we design 3 level tasks: **Easy** (SCV and BattleCruiser), **Medium** (SCV, Thor, Banshee, and Raven), **Hard** (SCV, SiegeTank, Medivac, VikingFighter, and Ghost).

The primary factor contributing to the escalation in task difficulty is the increase in the number of unit types. Besides, each task comprises a variety of instructions, differentiated by distinct unit quan-

tity combinations. For example, in Medium task, the instructions "(16 SCVs, 3 Thors, 3 Banshees, 4 Ravens)" and "(22 SCVs, 4 Thors, 3 Banshees, 5 Ravens)" exemplify this diversity.

### C.2.2 REWARD FUNCTION

The following python program details the reward function for StarCraft II Unit Building.

```python
############### Reward Function for StarCraft II Unit Building ###############

def building_ins_reward(self, obs, next_obs, parsed_ins):
    # Reward for building construction: the increase in the number of the units to be trained
    reward = 0
    negative_reward_scale = -1
    # 1. the increase in the number of the units to be trained
    if isinstance(parsed_ins, dict):
        units = parsed_ins.keys()
    elif isinstance(parsed_ins, list):
        # units = parsed_ins
        raise NotImplementedError("parsed_ins should be a dict, not a list")

    for k in units:
        # reward on the change of the number of the units to be trained
        c_k_obs, u_k_obs = self.obtain_unit_count(obs, k.upper())
        c_k_next_obs, u_k_next_obs = self.obtain_unit_count(next_obs, k.upper())

        k_obs = c_k_obs + u_k_obs
        k_next_obs = c_k_next_obs + u_k_next_obs

        if k_obs >= parsed_ins.get(k):
            # negative reward for the units that have been trained enough
            reward += negative_reward_scale * (k_next_obs - k_obs)
        else:
            if k_next_obs <= parsed_ins.get(k):
                reward += k_next_obs - k_obs
            else:
                reward += ((parsed_ins.get(k) - k_obs) + negative_reward_scale * (k_next_obs -
                ↪    parsed_ins.get(k)))

    return reward
```

# D EXPERIMENTAL SETTINGS DETAILS

## D.1 FAIRNESS EXPLANATION

We quantify the number of tasks that the learning agent of each method must traverse in ALFWorld. The results demonstrate that the number of tasks CoPiC needs to learn is comparable to that of Reflexion and AdaPlanner. This finding validates the fairness of the comparison between CoPiC and the baselines.

**ALFWorld** is a text-based virtual household environment featuring six distinct task types: Pick and Place (Pick), Examine in Light (Examine), Clean and Place (Clean), Heat and Place (Heat), Cool and Place (Cool), and Pick Two and Place (Pick Two). Each task type consists of a training task set, a seen task set, and an unseen task set. Tasks in the seen task set consist of known task instances {task-type, object, receptacle, room} that appear in the training set. Tasks in the unseen task set consist of new task instances that do not appear in the training task set. The number of tasks in each set is as follows:

| Task Type | train | seen | unseen |
|---|---|---|---|
| Pick | 790 | 35 | 24 |
| Examine | 308 | 13 | 18 |
| Clean | 650 | 27 | 31 |
| Heat | 459 | 16 | 23 |
| Cool | 533 | 25 | 21 |
| Pick Two | 813 | 24 | 17 |
| All | 3553 | 140 | 134 |

Table 4: Number of tasks in each set of ALFWorld

Reflexion and AdaPlanner both learn directly from unseen test tasks. Reflexion is a framework that reinforces language agents with a "trial-and-error" mechanism, repeating the process for each task. AdaPlanner is a closed-loop planning method that generates and adjusts programs iteratively to interact with each task. Therefore, both methods use the total 134 tasks from the unseen test set during the learning process.

For CoPiC, we **(1) learn from several training tasks and then deploy the learned planning programs and selector to unseen test tasks without incurring additional LLM querying costs or further selector fine-tuning**. For each type of task, CoPiC selects 20 tasks randomly from the training set for evolving planning programs and fine-tuning the selector. Therefore, **(2) the total number of tasks used for the six types of tasks during CoPiC's learning is $20 \times 6 = 120$, which is similar to the 134 tasks used in Reflexion and AdaPlanner**. Notably, since fine-tuning our selector is essentially a learning process for a neural network, we believe that learning on the training set and then evaluating on the unseen test set constitutes a more appropriate setup.

We emphasize that, **compared to the experimental setup of the baselines, CoPiC's setup imposes higher demands on the model, requiring it to generalize zero-shot to unseen tasks. Therefore, CoPiC outperforms the baselines under more stringent setup**. Besides, CoPiC can also perform online learning on the test set same as baselines. Here we constructed such setup identical to that of the baselines by allowing CoPiC to perform online learning on the test set, namely CoPiC(TSL) in Table 1. Under this completely consistent setup, CoPiC(TSL) still achieved a success rate that was 17.17% higher than the baselines, while reducing token consumption by 87.16%.

## D.2 DETAILED HYPERPARAMETERS

We detailed the hyperparameters appearing in Algorithm 1 of CoPiC in Table 5.

| Hyperparameters | Value |
|---|---|
| $n$ | 3 |
| $N$ | 20 |
| $\max\_Trials$ | 10 |
| $M$ | 5 |
| $K$ | 128 |

Table 5: The detailed hyperparameters used in CoPiC.

# E  ADDITIONAL EXPERIMENTAL RESULTS

## E.1  STARCRAFT II UNIT BUILDING

Here, we present the experimental results for the **Easy** and **Medium** tasks of the StarCraft II Unit Building, as detailed in Table 6. In the Easy task, CoPiC achieved the same success rate as advanced baselines at a 88.48% cost reduction. In the Medium task, CoPiC's success rate was 35.50% higher than that of advanced baselines, while reducing the cost by 75.97%.

| Method | Easy | | Medium | |
|---|---|---|---|---|
| | SR ↑ | Cost ↓ | SR ↑ | Cost ↓ |
| CoPiC (Ours) | **100.00** | **0.07M** | **100.00** | **0.17M** |
| AdaPlanner | **100.00** | 0.41M | 64.00 | 0.53M |
| Reflexion | **100.00** | 0.89M | 64.00 | 0.98M |
| Prospector | **100.00** | 0.91M | 71.00 | 0.95M |
| REPL-Plan | **100.00** | 0.52M | 59.00 | 0.59M |
| GPT-3.5 | 32.00 | 0.56M | 0.00 | 0.6M |
| Cot-Zero-Shot | 20.00 | 0.6M | 0.00 | 0.63M |
| Cot-Few-Shot | **100.00** | 0.57M | 50.00 | 0.65M |
| PPO | 0.00 | – | 0.00 | – |

Table 6: Results on the **Easy** and **Medium** tasks of StarCraft II Unit Building. SR denotes Success Rate and Cost denotes LLMs Querying Costs. **Step** in this Table means the metric **Interact Steps**

## E.2  COMPARSION OF TOKEN COSTS BETWEEN PLANNER AND SELECTOR IN COPIC

| LLMs | Module | Input Tokens | Output Tokens | Total Tokens |
|---|---|---|---|---|
| GPT-3.5 | LLMs Query | 0.40M | 0.05M | 0.45M |
| | TinyLlama Query | 0.55M | 0.02M | 0.57M |
| DeepSeek-Coder | LLMs Query | 0.14M | 0.02M | 0.16M |
| | TinyLlama Query | 0.63M | 0.03M | 0.66M |
| DeepSeek-V3 | LLMs Query | 0.18M | 0.03M | 0.21M |
| | TinyLlama Query | 0.65M | 0.03M | 0.68M |
| Qwen2.5-14B | LLMs Query | 0.11M | 0.02M | 0.13M |
| | TinyLlama Query | 0.64M | 0.03M | 0.67M |

Table 7: Comparison on total tokens costs between LLMs querying and TinyLlama querying (selector) in CoPiC on ALFWorld. M is million.

In CoPiC, we compared the token consumption of LLMs in the Planner for generating and enhancing planning programs with that of TinyLlama in the Selector for plan evaluation and fine - tuning, as presented in Table 7. Despite consuming 2.72× tokens of LLMs, the cost of querying TinyLlama (the selector) during training and testing is negligible due to its small size (1.1B), which is only 1/13 (Qwen2.5-14B) to 1/610 (DeepSeek-V3) of our LLMs.

### E.3 METRIC: STEP

**Step** refers to the number of environment interaction steps needed to complete tasks during testing, reflecting planning quality (fewer steps indicate higher quality). As shown in Table 8, CoPiC achieves a **30.43% reduction** in **Step** compared with advanced baselines, demonstrating its superior planning quality.

| Method | Pick | Examine | Clean | Heat | Cool | Pick Two |
|---|---|---|---|---|---|---|
| CoPiC (Ours) | 16.06 | **16.94** | **14.63** | **16.83** | **14.60** | **22.74** |
| AdaPlanner | **15.92** | 23.33 | 16.94 | 30.50 | 18.90 | 25.47 |
| Reflexion | 29.31 | 38.18 | 34.09 | 28.44 | 35.81 | 35.14 |
| Prospector | 18.10 | 24.33 | 21.29 | 26.26 | 21.71 | 26.67 |
| REPL-Plan | 21.11 | 18.29 | 22.21 | 23.14 | 21.37 | 27.15 |
| GPT-3.5 | 50.00 | 48.06 | 50.00 | 50.00 | 47.95 | 50.00 |
| Cot-Zero-Shot | 48.92 | 49.28 | 50.00 | 50.00 | 50.00 | 50.00 |
| Cot-Few-Shot | 45.29 | 45.00 | 47.35 | 43.74 | 43.90 | 45.06 |

Table 8: Comparison of CoPiC and baselines in ALFWorld on metric **Step** $\downarrow$.

### E.4 ABLATION ON THE NUMBER OF PLANNING PROGRAMS

We conducted an ablation study on the number of planning programs in the StarCraft II Unit Building environment. Specifically, we froze the learned selector in CoPiC, discarded the existing planning programs, and instructed the LLM to generate new planning programs guided by the frozen selector. The impact of varying the number of planning programs from 1 to 4 is summarized in Table 9. The results show that multiple planning programs consistently succeeded in completing all three task types, a feat unattainable by a single program. Thus, **utilizing multiple planning programs significantly outperforms relying on a single one**. Additionally, among configurations using 2 to 4 programs, the use of 3 programs achieved optimal performance at the lowest cost. This balance avoids both the underrepresentation of essential technology trees caused by too few programs and the increased complexity in the selector's evaluation with too many programs.

### E.5 STANDARD DEVIATION OF CoPiC VS. BASELINES IN ALFWORLD

Tabel 10 shows the standard deviation of the task success rate across 5 runs for each method in ALFWorld. Owing to the stability of the planning programs (which produce the same output given the same input) and the Selector's acquisition of domain-specific knowledge, CoPiC exhibits more stable performance than baselines.

### E.6 RESULTS OF PPO ON ALFWORLD

Table 11 shows the results of PPO in ALFWorld. We emphasize that we have thoroughly tuned the hyperparameters of PPO, including batch size $\in [32, 64, 128, 256]$, clip ratio $\in [0.1, 0.2, 0.3]$, policy learning rate $\in [5 \times 10^{-7}, 5 \times 10^{-5}, 1 \times 10^{-5}, 1 \times 10^{-4}, 3 \times 10^{-4}, 1 \times 10^{-3}]$, and entropy coefficient

| Task | Number | SR $\uparrow$ | Cost $\downarrow$ |
|---|---|---|---|
| Easy | n = 1 | 0.67 | 0.16M |
| | n = 2 | **1.00** | 0.10M |
| | n = 3 | **1.00** | **0.05M** |
| | n = 4 | **1.00** | 0.08M |
| Medium | n = 1 | 0.33 | 0.18M |
| | n = 2 | **1.00** | 0.21M |
| | n = 3 | **1.00** | **0.14M** |
| | n = 4 | **1.00** | 0.23M |
| Hard | n = 1 | 0.33 | 0.13M |
| | n = 2 | **1.00** | 0.11M |
| | n = 3 | **1.00** | **0.04M** |
| | n = 4 | **1.00** | 0.08M |

Table 9: Ablation on Planners in StarCraft II Unit Building. n is the number of planning programs.

| Method | Pick | Examine | Clean | Heat | Cool | Pick Two |
|--------|------|---------|-------|------|------|----------|
| CoPiC | 0.00 | 0.00 | 0.00 | 0.00 | 0.00 | 4.40 |
| AdaPlanner | 0.00 | 10.30 | 6.32 | 7.58 | 6.32 | 4.40 |
| Reflexion | 7.91 | 10.89 | 6.58 | 12.17 | 7.74 | 8.65 |
| Prospector | 7.45 | 11.33 | 6.64 | 6.15 | 10.17 | 12.56 |
| REPL-plan | 7.17 | 10.54 | 8.31 | 7.28 | 7.13 | 0.00 |
| GPT-3.5 | 3.33 | 9.30 | 4.08 | 6.74 | 3.01 | 5.76 |
| CoT-Zero-Shot | 3.73 | 6.09 | 5.62 | 10.43 | 6.02 | 13.10 |
| CoT-Few-Shot | 4.56 | 4.97 | 4.08 | 9.53 | 4.26 | 11.16 |

Table 10: Standard Deviation ($\downarrow$) of task success rate across 5 runs for each method

| Method | Pick | | Examine | | Clean | | Heat | | Cool | | Pick Two | |
|--------|------|------|---------|------|-------|------|------|------|------|------|----------|------|
| | SR $\uparrow$ | Cost $\downarrow$ | SR $\uparrow$ | Cost $\downarrow$ | SR $\uparrow$ | Cost $\downarrow$ | SR $\uparrow$ | Cost $\downarrow$ | SR $\uparrow$ | Cost $\downarrow$ | SR $\uparrow$ | Cost $\downarrow$ |
| CoPiC (Ours) | **100.00** | **0.05M** | **100.00** | **0.04M** | **100.00** | **0.33M** | **100.00** | **0.26M** | **100.00** | **0.28M** | 95.29 | **0.06M** |
| PPO | 0.00 | – | 0.00 | – | 0.00 | – | 0.00 | – | 0.00 | – | 0.00 | – |

Table 11: Comparison of CoPiC and PPO in ALFWorld. M represents millions.

$\in [0.001, 0.01, 0.1]$. The vast observation and action spaces, combined with out-of-distribution (OOD) test sets and sparse rewards, pose significant challenges. Consequently, PPO, without any prior knowledge, is unable to effectively handle the planning tasks in ALFWorld, resulting in a success rate that consistently remains at 0. Besides, Table 12 also listed other key hyperparameters of PPO used in the results.

### E.7 VALIDATING CoPiC'S SCALABILITY USING ROBOSUITE

In this section, we validated the scalability of CoPiC. We conducted experiments with an UR5e robotic arm equipped with a two-finger gripper in Robosuite (Zhu et al., 2020), **an environment with visual observation inputs and continuous action outputs**. The tasks we used include Lift, Pick-and-Place, and Stack-Three-Cubes. To extend CoPiC to Robosuite, we provided some necessary information about the environment, mainly the action space of the environment, which includes actions such as move_gripper(posX, posY, posZ), open_gripper(), close_gripper(), etc. In the following paragraphs, we first describe our workflow pipelines and then present the experimental results.

**Workflow Pipelines Overview**

- **Perception**: Utilize vision foundation models to transform RGB image inputs into text inputs.
- **Planning**: CoPiC completes high-level planning based on the text inputs.
- **Control**: Use low-level controllers to convert the high-level plans into executable actions in the environment to complete the tasks.

The details are as follows.

| Hyperparameter | Value |
|----------------|-------|
| epsilon | 0.2 |
| lambda (GAE) | 0.9 |
| critic learning rate | 1e-5 |
| mini batch size | 32 |
| epochs per update | 10 |
| value loss cofficient | 0.5 |
| gamma | 0.99 |

Table 12: Other key hyperparameters used in PPO.

| Method | Lift | Pick-and-Place | Stack-Three-Cubes |
|--------|------|----------------|-------------------|
| CoPiC | 100.00 | 96.67 | 91.67 |
| GPT-3.5 | 78.33 | 65.00 | 48.33 |

Table 13: Comparison of CoPiC and GPT-3.5 on Robosuite

**(1) Perception**: The input for the task is an RGB image. We first use GroundingDINO (Liu et al., 2024c) to process the image and obtain the bounding boxes for all objects within it. Then, we use SAM (Kirillov et al., 2023) to generate precise segmentation masks for the objects based on the image and bounding boxes. Next, we process the image with DepthPro (Bochkovskii et al., 2024) to obtain a depth map. By combining the camera intrinsic parameters, we back-project the segmentation masks of the objects from pixel space to 3D space to obtain 3D point clouds, and take the median of the point cloud as the position vector of the objects. We then describe the task in text format based on the position vectors, in a format similar to: "Move the cube located at (0, 1.1, 0) to the plate located at (2, 2.4, 0)".

**(2) Planning**: CoPiC generates planning programs based on the "task text description" and "the descriptions of action space" (including move_gripper(posX, posY, posZ), open_gripper(), close_gripper(), etc.) to perform planning.

**(3) Control**: We use the predefined low-level controllers (i.e pre-defined APIs of move_gripper(posX, posY, posZ), open_gripper(), close_gripper(), etc.) to ground the plan in the environment.

**Results**: For each type of task, during the test, we randomly generated 20 layouts for the objects based on a distribution different from that used in training. The success rate averaged over three seeds of CoPiC and GPT-3.5 are as follows. GPT-3.5 refers to using GPT-3.5 directly for planning in the Planning part of the workflow pipeline above (with the same action space information provided as for CoPiC). **As can be seen from the Table 13, CoPiC achieved a success rate that was 32.23% higher than that of GPT-3.5, demonstrating that CoPiC can be extended to tasks that require visual perception, planning, and sophisticated control simultaneously.**

### E.8 CoPiC is Robust on Temperature Variation

In this section, we discuss the sensitivity of CoPiC to variations in temperature settings. The following ablation studies on different temperatures demonstrate that temperature has a certain impact on CoPiC's performance and cost, but CoPiC is not highly sensitive to temperature variations. **Overall, CoPiC can adapt a broad range of temperature settings ($0.2 \sim 0.8$), thereby ensuring both the diversity and quality of the plans**.

| Method | Pick | | Examine | | Clean | | Heat | | Cool | | Pick Two | |
|--------|------|------|---------|------|-------|------|------|------|------|------|----------|------|
| | SR ↑ | Cost ↓ | SR ↑ | Cost ↓ | SR ↑ | Cost ↓ | SR ↑ | Cost ↓ | SR ↑ | Cost ↓ | SR ↑ | Cost ↓ |
| CoPiC (0.2) | 100.00 | 0.12M | 90.00 | 0.08M | 92.90 | 0.63M | 97.37 | 0.57M | 95.24 | 0.38M | 92.94 | 0.11M |
| CoPiC (0.5) | 100.00 | 0.13M | 95.56 | 0.07M | 91.61 | 0.55M | 96.52 | 0.48M | 100.00 | 0.37M | 96.47 | 0.09M |
| CoPiC (0.6) | 100.00 | 0.05M | 100.00 | 0.04M | 100.00 | 0.33M | 100.00 | 0.26M | 100.00 | 0.28M | 95.29 | 0.06M |
| CoPiC (0.8) | 100.00 | 0.10M | 100.00 | 0.07M | 99.35 | 0.39M | 94.78 | 0.52M | 91.43 | 0.29M | 97.65 | 0.07M |

Table 14: The impact of temperature variations on CoPiC's performance. CoPiC(0.2) denotes setting the temperature of the LLM to 0.2, with the same logic applying to the other rows.

### E.9 CoPiC can Benefit from Experiences of Other Type of Tasks.

In this section, we **compared the performance of CoPiC's planning programs initialized with and without interaction experiences** from the Pick task on the Clean, Heat, and Cool tasks in ALF-World. The results are presented in Table 15. It can be observed that with interaction experiences from previously completed tasks incorporated as part of the prompt, the success rate of CoPiC's initially generated planning programs increased by 11.63%. This indicates that **LLM can enhance its capability to generate initial plans by leveraging existing experiences**.

| Method | Clean | Heat | Cool |
|---|---|---|---|
| CoPiC w Pick Exp | 43.87 | 53.91 | 14.29 |
| CoPiC w/o Pick Exp | 38.06 | 39.13 | 0.00 |

Table 15: Comparison of CoPiC with and without interaction experiences from Pick tasks.

| Task | GPT-3.5 | DeepSeek-Coder | DeepSeek-V3 | Qwen2.5-Coder-14B-Instruct | Qwen2.5-14B-Instruct |
|---|---|---|---|---|---|
| Pick | 0.2153 (0.2682) | 0.3516 (0.4167) | 0.3455 (0.3750) | 0.4193 (0.5208) | 0.2023 (0.3672) |
| Examine | 0.3342 (0.4062) | 0.3177(0.4453) | 0.3681 (0.3776) | 0.3741 (0.3229) | 0.3741 (0.4453) |

Table 16: Quantification of Diversity

### E.10 QUANTIFICATION OF DIVERSITY ON A SET OF PLANNING PROGRAMS

In this section, we present the quantification of diversity on a set of planning programs. When generating planning programs, we set a relatively high temperature (0.6) for the LLMs and employed multiple sampling to produce these programs. **Given the large planning space of the tasks and the lack of domain-specific experience in general-purpose LLMs**, the programs generated through multiple sampling exhibit sufficient diversity. Table 16 presents the quantitative diversity results for a set of planning programs generated by each base model on the Pick and Examine tasks in ALFWorld. The calculation formula is:

$$\text{Diversity} = 1 - \frac{2}{N(N-1)} \sum_{i=1}^{N-1} \sum_{j=i+1}^{N} \text{sim}(\rho_i, \rho_j) \tag{10}$$

where $\text{sim}(\rho_i, \rho_j)$ denotes the MinHash Jaccard Similarity between two planning programs. The results in the table, with each entry formatted as **Mean Diversity (Max Diversity)** for three sets of planning programs, each set comprising three programs generated by the LLM for the task, demonstrate that **the planning programs generated by CoPiC exhibit good diversity**.

For the same task, the planning programs generated by the LLM may share the same overall logic. For instance, the planning logic for the Cool task might consistently be "find object → pick object → cool object → find receptacle → place object". **The actual differences often manifest in the specific implementation of a particular step**, such as variations in the logic for "finding object". Below are two concrete examples, with **Diversity = 0.2813**. It can be observed that when the object is not found, **the first planning program will conduct simple exploration**, while **the second planning program will attempt to first traverse the receptacles that may contain the object, resorting to exploration only if no available receptacles are found**. It can be observed that when the object is not found, **the first planning program will conduct simple exploration**, while **the second planning program will attempt to first traverse the receptacles that may contain the object, resorting to exploration only if no available receptacles are found**. Due to the complexity of the tasks, such differences in logic will exist among different programs.

```python
def pick_cool_then_place(self, objecttype, receptacletype):
    # Step 1: Find and Pick the object
    if self.name2type(self.holding) != self.name2type(objecttype):
        # Find the object
        entity = self.find_object(objecttype)
        if entity is None:
            # Explore the environment to find the object
            self.explore()
            return
        else:
            # Go to the object's receptacle and take it
            self.goto(entity.in_on)
            r_entity = self.seen_entitys[entity.in_on]
            if r_entity.openable and not r_entity.isopen:
                self.open_receptacle(r_entity.name)
            self.take(entity.name, entity.in_on)
            return
    else:
```

```
        # Step 2: Cool the object in a fridge
        ...
        # Step 3: Find specified receptacle and Place the cooled object in it
        ...
```

```python
def pick_cool_then_place(self, objecttype, receptacletype):
    # Step 1: Find and Pick the object
    if self.name2type(self.holding) != self.name2type(objecttype):
        # Find the object
        entity = self.find_object(objecttype)
        if entity is None:
            # Find receptacletypes that can contain the object
            r_types = self.find_canbe_contained(objecttype)
            if r_types is None:
                # Explore the environment to find the object
                self.explore()
                return
            else:
                # Check each receptacletype to find the object
                for r_type in r_types:
                    r_entitys = self.find_receptacles(r_type)
                    if r_entitys is None:
                        continue
                    for r_entity in r_entitys:
                        self.goto(r_entity.name)
                        if r_entity.openable and not r_entity.isopen:
                            self.open_receptacle(r_entity.name)
                        # Check if the object is inside the receptacle
                        if entity is None:
                            entity = self.find_object(objecttype)
                        if entity is not None:
                            break
                    if entity is not None:
                        break
                if entity is None:
                    # Explore the environment to find the object
                    self.explore()
                    return
        # Go to the object's receptacle and take it
        ...
    else:
        # Step 2: Cool the object in a fridge
        ...
        # Step 3: Find specified receptacle and Place the cooled object in it
        ...
```

### E.11 ANALYSIS OF NUMBER OF PLANNING PROGRAMS ON SUPERHARD TASK

| n | 2 | 3 | 4 | 5 | 6 |
|---|---|---|---|---|---|
| SR ↑ | 32.00 | 44.00 | 72.00 | 96.00 | 92.00 |
| Cost ↓ | 1.71 | 2.14 | 3.92 | 2.42 | 3.88 |

Table 17: Results for varying $n$ on SuperHard Task.

We further constructed the **SuperHard** task on the StarCraft II Unit Building environment, which requires building all 17 Terran unit types. The results for varying values of $n$ on this task are shown in Table 17. It can be seen that on the SuperHard task, the number of required planning programs has also increased to 5. The performance of the $n = 6$ slightly drops compared with $n = 5$, because a larger n raises the difficulty for the selector to pick the plan most aligned with the long-term reward—a trade-off between the number of programs and performance. Overall, the greater variety of unit types implies a more complex tech tree, and to fully construct this complex tech tree, more planning programs are needed to work in coordination. And CoPiC's diverse planning programs, combined with the domain-adaptive selector, enable it to scale and tackle such complex tasks.

### E.12 ANALYSIS OF THE SELECTOR'S FREQUENCY IN SWITCHING PLANNING PROGRAMS

| Task | Pick | Examine | Clean | Heat | Cool | Pick Two |
|---|---|---|---|---|---|---|
| Switch Interval | 2.78 | 1.52 | 1.34 | 1.68 | 1.76 | 2.32 |

Table 18: The intervals at which the Selector switches planning programs.

The Table 18 shows the intervals at which the Domain-Adaptive Selector switches planning programs to generate plans in ALFWorld. The results indicate that for simpler tasks, such as Pick, the switching intervals are longer (i.e., lower frequency), while for more complex tasks, such as Examine and Clean, the switching intervals are shorter (i.e., higher frequency). This finding suggests that as task complexity increases, the number of necessary planning programs correspondingly rises. Consequently, the selector needs to switch plans more frequently in these more challenging tasks to adapt to their rising complexity. This also shows that generating a single planning program capable of completing the task is difficult, whereas CoPiC produces diverse planning programs and, with the domain-adaptive selector, accomplishes the task at a lower cost.

### E.13 COPIC'S 'MINIMAL-EXAMPLE' EXPERIMENT

| Method | Medium | | Hard | | SuperHard | |
|---|---|---|---|---|---|---|
| | SR ↑ | Cost ↓ | SR ↑ | Cost ↓ | SR ↑ | Cost ↓ |
| CoPiC (minimal-example) | **100.00** | **0.18M** | **100.00** | **0.08M** | **92.00** | **2.45M** |
| CoPiC | **100.00** | **0.17M** | **100.00** | **0.06M** | **96.00** | **2.42M** |
| AdaPlanner | 64.00 | 0.53M | 71.00 | 0.36M | 32.00 | 6.08M |
| Reflexion | 64.00 | 0.98M | 24.00 | 0.78M | 34.00 | 6.74M |
| Prospector | 71.00 | 0.95M | 68.00 | 0.68M | 42.00 | 7.02M |
| REPL-Plan | 59.00 | 0.59M | 58.00 | 0.39M | 39.00 | 5.61M |

Table 19: 'Minimal-Example' Results on the Medium, Hard and SuperHard tasks of StarCraft II Unit Building. SR denotes Success Rate and Cost denotes LLMs Querying Costs.

In this section, we conducted an additional 'minimal-example' experiment on StarCraft II Unit Building (SC2) to demonstrate that CoPiC's high performance should not attributable to the effort put into initial prompts, but from its core design of the iterative refinement of planning programs and domain-adaptive selector. Specifically, we replaced the planning program example in CoPiC's prompt with the simplest possible SC2 task – Building SCV (i.e. the example provided in Sec 4.1.1) – thereby minimizing the environmental information conveyed by the prompt. The results are summarized in the Table 19. As we can see, (1) CoPiC(minimal-example) is almost **identical** to

CoPiC in both success rate (97.33% vs. 98.67%) and cost (0.90M vs. 0.88M); (2) CoPiC(minimal-example) still achieves a **45.17%** higher success rate and **64.70%** lower cost than the baselines. This outcome confirms that CoPiC's high performance is rooted in its architectural innovations, not in prompt engineering.

# F PROGRAMS EVOLUTION PROCESS

In CoPiC, LLMs do not indiscriminately improve programs but instead **analyze interaction histories** to make targeted enhancements. We provide an example to illustrate this.

**Old Interaction History (only key parts)**

```
Obs: You are in the middle of a room. Looking quickly around you, you see ...
Your task is to: cool some egg and put it in microwave.
...
Act: go to countertop 1
Obs: On the countertop 1, you see a apple 1, a creditcard 1, a egg 1, a fork 2, a knife 2, a peppershaker
↪  1, a plate 1, and a spoon 1.
Act: go to countertop 1 # Redundant action (comment just for explaining, not included in prompt)
Obs: Nothing happens. # No state change (comment just for explaining, not included in prompt)
...
```

The agent had already reached `countertop 1` and identified `egg 1`. The logical next action should be `take egg 1`, but the program redundantly re-executed `go to countertop 1`.

**Original Planning Program (with Flaw)**

```python
def pick_cool_then_place(self, objecttype, receptacletype):
    # First, pick up the object if we're not already holding it
    if self.name2type(self.holding) != self.name2type(objecttype):
        # Find the object
        ...
                # Check each receptacletype to find the object
                for r_type in r_types:
                    # find and check each receptacle of receptacletype
                    r_entitys = self.find_receptacles(r_type)
                    if r_entitys is None:
                        continue
                    for r_entity in r_entitys:
                        self.goto(r_entity.name)
                        if r_entity.openable and not r_entity.isopen:
                            self.open_receptacle(r_entity.name)
                            return
    else:
        # We're holding the object - now we need to cool it
        ...
        # After cooling, place the object in the target receptacle
        ...
```

Based on the interaction history, the LLM identified issue: the program navigated (`self.goto(r_entity.name)`) to receptacles and opened (`self.open_receptacle(r_entity.name)`) them but failed to take the target object (egg 1 in this case). The `take` action was entirely absent from the logic flow.

**LLM-Enhanced Program**

```python
def pick_cool_then_place(self, objecttype, receptacletype):
    # First check if we're already holding the object we need to cool
    if self.name2type(self.holding) != self.name2type(objecttype):
        # Find the object
        ...
                # Check each receptacletype to find the object
                ... # same as the old program
                    # New critical addition +++++++++++++
                    # Check if object is directly on this receptacle
                    if hasattr(r_entity, 'contents'):
                        for obj in r_entity.contents:
                            if self.name2type(obj.name) == self.name2type(objecttype):
                                self.take(obj.name, r_entity.name)
                                return
    else:
        # We have the object, now we need to cool it
        # Find a fridge to cool the object
        ...

        # Now place the cooled object in the target receptacle
        ...
```

Key Improvement: Added explicit checks for object presence and a `self.take()` call to take the target item after navigation.

**New Interaction History (only key parts)**

```
Obs: You are in the middle of a room. Looking quickly around you, you see ...
Your task is to: cool some egg and put it in microwave.
...
Act: go to countertop 1
Obs: On the countertop 1, you see a apple 1, a creditcard 1, a egg 1, a fork 2, a knife 2, a peppershaker
↪   1, a plate 1, and a spoon 1.
Act: take egg 1 from countertop 1  # useful action
Obs: You pick up the egg 1 from the countertop 1.
...
```

We emphasize that while current LLM-based program improvement methods remain probabilistic, CoPiC's analysis of interaction histories provides two critical guarantees: (1) Problem Diagnosis: Failures explicitly expose flawed logic (e.g., missing `take` actions), enabling targeted corrections. (2) Measurable Progress: Enhancements to planning programs manifest either through increased task success rates or the elimination of observed failure modes (e.g., redundant navigation due to missing `take` logic).

## G  PSEUDOCODE OF COPIC

---

**Algorithm 1 CoPiC**

---

**Input:** Number of planning programs $n$, episodes $N$, total Trials $max\_Trials$, summary size $M$,
    **init_prompt**, **evolve_prompt**, selector parameters $\theta$, selector fine-tuning steps $K$
**Output:** Planning programs $\{\rho_i\}_{i=1}^n$, selector $C_\theta$
 1: Initialize $n, N, M, max\_Trials,$ **init_prompt**, **evolve_prompt**, $\theta, K$
 2: Initialize interaction history $H \leftarrow \{\}$, replay buffer $D \leftarrow \{\}$, environment step $step \leftarrow 0$
 3: Initialize success rate threshold **threshold**
 4: Initialize $\{\rho_i\}_{i=1}^n$ using **init_prompt**
 5: Set cur$\_Trials \leftarrow 0$
 6: **while** True **do**
 7:     Set $n\_success \leftarrow 0$
 8:     **for** episode = 1 to $N$ **do**
 9:         Reset env, get instruction $I$, observation $o$
10:         **while** not $done$ **do**
11:             Generate candidate plans $\{p_i = \rho_i(p_i|I, o)\}_{i=1}^n$
12:             Select plan $p = C_\theta(I, o, \{p_i\}_{i=1}^n)$
13:             Step environment with plan $p$, receive $o'$, reward $r$, success flag $signal$, done flag $done$
14:             Store $(I, o, p, r, o', done)$ in buffer $D$
15:             Store $(I, o, p, signal)$ in history $H$
16:             $n\_success \leftarrow n\_success + signal$
17:             **if** step % $K$ == 0 **then**
18:                 Fine-tune selector $\theta$ using buffer $D$ via PPO objective shown in Eq 11
19:                 Reset buffer $D$
20:             **end if**
21:             $o \leftarrow o'$
22:         **end while**
23:     **end for**
24:     **if** $n\_success/N \geq$ **threshold then**
25:         **break**
26:     **end if**
27:     cur$\_Trials \leftarrow$ cur$\_Trials + 1$
28:     **if** cur$\_Trials \geq$ max$\_$Trials **then**
29:         **break**
30:     **end if**
31:     Summarize the last $M$ episodes in $H$ as an interaction summary
32:     Evolve new planning programs $\{\rho_i\}_{i=1}^n$ using **evolve_prompt** and interaction summary
33:     Reset history $H$
34: **end while**
35: **Return** $\{\rho_i\}_{i=1}^n, C_\theta$

---

The PPO objective used for fine-tuning Selector is:

$$\mathcal{L}^{PPO}(\theta) = \mathbf{E}_t[\min(r_t(\theta)\hat{A}_t, \text{clip}(r_t(\theta), 1-\epsilon, 1+\epsilon)\hat{A}_t) - c_1(V_\theta(s_t) - R_t)^2 + c_2 H[C_\theta](o_t)] \quad (11)$$

where $r_t(\theta) = \frac{C_\theta(p_t|o_t, I)}{C_{\theta_{old}}(p_t|o_t, I)}$ is the probability ratio; $\hat{A}_t$ is an estimator of the advantage function; $V_\theta$ is the value function; $H[C_\theta](o_t)$ is the entropy bonus.

# H PROMPT DETAILS

In this section, we detail the prompts of CoPiC used in ALFWorld, Nethack, and StarCraft II Unit Building.

## H.1 PROMPTS IN ALFWORLD

### H.1.1 INITIALIZATION-ALFWORLD

```python
# You are a household agent. Here is some Python code defining a household environment:

# Entity class an object/receptacle in the environment, including its properties
class Entity:
    def __init__(self, **kwargs):
        self.name = None
        self.loc = None # location
        self.in_on = None # the receptacle that the object is in/on
        self.ishot, self.iscool, self.isclean = None, None, None
        self.isopen, self.ison, self.istoggled = None, None, None
        self.pickupable, self.openable, self.toggleable = None, None, None
        self.heatable, self.coolable, self.cleanable = None, None, None
        self.isobject, self.isreceptacle, self.isreceptacleobject = None, None, None
        self.type = None
        self.checked = None
        for key, value in kwargs.items():
            setattr(self, key, value)

        assert self.name is not None
        assert self.type is not None

# Entitys class stores all the entities in the environment
class Entitys:
    # return the entity with the given name, where the format of the given name is similar to "apple 22".
    def __getitem__(self, entity_name):
        ...

# Agent class represents the state of the agent, including its location,
# what it's holding, entities it has seen, as well as the actions it can take.
class Agent:
    def __init__(self, ...):
        self.holding = None
        self.location = None
        self.seen_entitys = Entitys()

    # Here are some assistant methods the agent can using:

    # return the receptacletypes that can contain the objecttype
    def find_canbe_contained(self, objecttype: str):
        ...

    # return an object(Entity) with the given objecttype
    def find_object(self, objecttype: str):
        ...

    # return a list of object(Entity) with the given objecttype
    def find_objects(self, objecttype: str):
        ...

    # return a receptacle(Entity) with the given receptacletype
    def find_receptacle(self, receptacletype: str):
        ...

    # return a list of receptacle(Entity) with the given receptacletype
    def find_receptacles(self, receptacletype: str):
        ...

    # name2type transforms a name to a type, like 'apple', 'apple 22' or 'appletype ' -> 'appletype '
    def name2type(self, name: str):
        ...

    # Here are the admissible actions the agent can take:

    # Explore the environment and observe the entities in it.
    def explore(self):
        ...

    # Go to a receptacle
    # For example: goto('countertop 1').
```

```
    # Only goto(receptacle.name) and goto(object.in_on) is allowed. goto(entity.loc) is prohibited.
    def goto(self, receptacle):
        ...

    # Take an object from a receptacle if the agent is not holding anything.
    # For example: take('soapbar 1', 'towelholder 1')
    def take(self, object, receptacle):
        ...

    # Put an object in or on a receptacle if the agent is holding it.
    # For example: put('soapbar 1', 'cabinet 1')
    def put(self, object, receptacle):
        ...

    # Open a receptacle and observe its contents.
    # For example: open_receptacle('cabinet 1')
    def open_receptacle(self, receptacle):
        ...

    # Close an opened receptacle.
    # For example: close_receptacle('cabinet 1')
    def close_receptacle(self, receptacle):
        ...

    # Clean an object with sinkbasin.
    # For example: clean('soapbar 1', 'sinkbasin 1')
    def clean(self, object, receptacle):
        ...

    # Heat an object with a receptacle.
    # For example: heat('tomato 1', 'microwave 1')
    def heat(self, object, receptacle):
        ...

    # Cool an object with a receptacle.
    # For example: cool('pan 2', 'fridge 1')
    def cool(self, object, receptacle):
        ...

    # Turn on an object.
    # For example: turn_on('desklamp 1')
    def turn_on(self, object):
        ...

    # Method need to be completed for the task: <task>
    <task_method>:
        ...
# Now complete the `<task_method>` to solve the task by composing the agent's methods to interact with
↪  the environment.

# Here is an successful example of a solution to another type of task:

<example>

# Here is the actual task.
# <task>, that is, <task_method_desc>
# Referring to the successful example and its code, you should complete your solution function below:
<task_method>:
# Note: Do not directly call the `<example_method_name>` in the example. You should use its code as a
↪  reference.
```

## H.1.2 EVOLUTION-ALFWORLD

```
<Init_prompt>

# Here is an example of a solution to the task:

<example>

# Here is the actual task.
# <task>, that is, <task_method_desc>

You have generated code of <task_method> to solve the task as follows:
<python_program_0>

<python_program_1>

<python_program_2>
```

```
However, you executed the <task_method> function and get an interaction history as follows:
<interaction_history>

Let's think step by step. Referring to the successful case and the interaction history, you should
↪   generate superior solution function.

<task_method>:
# Note: Do not directly call the `<example_method_name>` in the example. You should use its code as a
↪   reference.
```

## H.2 PROMPTS IN NETHACK

### H.2.1 INITIALIZATION-NETHACK

```
You are an agent who plays the rogue-like game NetHack 3.6.6 using a `Python` program.
Here is a template for the `Python` program that interacts with NetHack, and you need to fill in the
↪   template at the placeholders (i.e., `init`, `update_init` and `core_function`) to accomplish a
↪   certain task in Nethack.

Here is the template:
```python
from netplay.core.skill_repository import SkillRepository
from netplay.nethack_agent.agent import NetHackAgent
from netplay.nethack_agent.skill_selection import *
from netplay.copic_agent.data import object_names, weapon_names, armor_names, \
    ring_names, amulet_names, tool_names, container_names, weptool_names, food_names, \
        potion_names, scroll_names, spell_names, wand_names, coin_names, gem_names, \
            rock_names, misc_names, questitem_names, object_can_pickup_names

class NethackTemplate(SimpleSkillSelector):

    def init(self, agent, dict_obs: dict):
        """
            placeholder: Add variables for this specific skill selector...
        """

    def update_init(self, agent, dict_obs: dict) -> dict:
        """
            placeholder: Update variables for this specific skill selector...
        """

    def core_function(self, agent, dict_obs: dict) -> SkillSelection:
        """
            placeholder: Fill in your core function here, using to interact with NetHack.
        """

```

Here is an example of `init`, `update_init` and `core_function` for accomplishing the task 'Find an
↪   item'.

<example>

Now fill the template to accomplish the tasks 'Upgrade your experience level to 3 in the 1st depth' in
↪   Nethack.
```

### H.2.2 EVOLUTION-NETHACK

```
<Init_prompt>
Now there are 3 `Python` programs used to interact with Nethack to accomplish the tasks 'Upgrade your
↪   experience level to 3 in the 1st depth':
Program 1:
```python
{$python_program_0$}
```
Program 2:
```python
{$python_program_1$}
```
Program 3:
```python
{$python_program_2$}
```
At each step, each program provides a plan. Subsequently, an oracle scoring model selects the optimal
↪   plan from the 3 plans to interact with the environment.
```

```
And the results of the 3 `Python` programs are:
{$the_interaction_results$}

With the 3 reference programs and their results, I need you explore and develop a more optimal program to
↪   accomplish the task. In addition, the class in the optimized program still need to be named
↪   `NethackTemplate`.
```

## H.3  PROMPTS IN STARCRAFT II UNIT BUILDING

### H.3.1  INITIALIZATION-STARCRAFT II UNIT BUILDING

```
You are an AI capable of generating `Python` programs to accomplish certain tasks in StarCraft II, and
↪   you have an in-depth understanding of all the knowledge about the Terran race in StarCraft II.

Here is an example for you to refer to on how to generate a program to accomplish the task:
<example>

The logic of the program's operation is to iteratively generate plans and interact with the game,
↪   ultimately completing the task.

Now your `task` is:
<task>
with the goal of training the specified quantities of the corresponding type of units in the game.

Now generate a program to accomplish this task. Your program should retain the comments from the program
↪   in the example. And your program should start with "```python" and end with "```". The function name
↪   in your program should be `planner`, with parameters `(obs, action_space, task)`.
```

### H.3.2  EVOLUTION-STARCRAFT II UNIT BUILDING

```
<Init_prompt>
Now there are 3 `Python` programs used to interact with the environment:
Program 1:
```python
{$python_program_0$}
```
Program 2:
```python
{$python_program_1$}
```
Program 3:
```python
{$python_program_2$}
```
At each step, each program provides a plan. Subsequently, an Oracle scoring model selects the optimal
↪   plan from the 3 plans to interact with the environment.

And the results of the 3 `Python` programs interacting with the environment is:
{$the_interaction_results$}

With the 3 reference programs and their result, I need you explore and develop a more optimal program to
↪   accomplish the task.

Your program should retain the comments from the program in the example. And your program should start
↪   with "```python" and end with "```". The function name in your program should be `planner`, with
↪   parameters `(obs, action_space, task)`.
```

# I  PLANNING PROGRAMS FROM COPIC

In this section, we provide examples of planning programs evolving from CoPiC in ALFWorld, NetHack, and StarCraft II Unit Building (hereafter referred to as SC2 for simplicity). To avoid overly long chapters, we present only the planning programs for the "pick and place", "upgrade exp level to 3", and "Hard (SCV, SiegeTank, Medivac, VikingFighter, Ghost)" tasks in the three environments, respectively. Planning programs for other tasks in each environment can be found in the supplementary material.

## I.1  PLANNING PROGRAMS IN ALFWORLD

### I.1.1  PICK AND PLACE

```python
############### program 1 ###############
def pick_and_place(self, objecttype, receptacletype):
    # Check if the agent is already holding an object
    if self.holding is not None:
        # If holding an object, put it down first
        self.put(self.holding, self.location)

    # Pick the object with the given objecttype
    if self.name2type(self.holding) != self.name2type(objecttype):
        # Find the object
        entity = self.find_object(objecttype)
        if entity is None:
            # Find receptacletypes that can contain the object
            r_types = self.find_canbe_contained(objecttype)
            if r_types is None:
                # Explore the environment
                self.explore()
                return
            else:
                # Check each receptacletype for the object
                for r_type in r_types:
                    r_entitys = self.find_receptacles(r_type)
                    if r_entitys is None:
                        continue
                    for r_entity in r_entitys:
                        self.goto(r_entity.name)
                        if r_entity.openable and not r_entity.isopen:
                            self.open_receptacle(r_entity.name)
                            return

    # Place the object in the receptacle
    r_entity = self.find_receptacle(receptacletype)
    if r_entity is not None:
        self.goto(r_entity.name)
        if r_entity.openable and not r_entity.isopen:
            self.open_receptacle(r_entity.name)
        self.put(objecttype, r_entity.name)
    else:
        # If the receptacle is not found, explore the environment
        self.explore()

############### program 2 ###############
def pick_and_place(self, objecttype, receptacletype):
    # Find an object of the given type
    object_entity = self.find_object(objecttype)
    if object_entity is None:
        # If the object is not found, explore the environment
        self.explore()
        return

    # Go to the receptacle that contains the object
    self.goto(object_entity.in_on)
    receptacle_entity = self.seen_entitys[object_entity.in_on]

    # If the receptacle is not open, open it
    if receptacle_entity.openable and not receptacle_entity.isopen:
        self.open_receptacle(receptacle_entity.name)

    # Take the object
    self.take(object_entity.name, object_entity.in_on)

    # Find the receptacle where the object will be placed
    receptacle_entities = self.find_receptacles(receptacletype)
```

```python
    if receptacle_entities is None:
        # If the receptacle type is not found, explore the environment
        self.explore()
        return

    # Go to the receptacle where the object will be placed
    self.goto(receptacle_entities[0].name)

    # Put the object in the receptacle
    self.put(object_entity.name, receptacle_entities[0].name)

############### program 3 ###############
def pick_and_place(self, objecttype, receptacletype):
    # Find the object of the given type
    object_entity = self.find_object(objecttype)

    if object_entity is None:
        # If object not found, explore the environment
        self.explore()
        return

    # Go to the receptacle type
    receptacle_entities = self.find_receptacles(receptacletype)

    if receptacle_entities is None:
        # If receptacle not found, explore the environment
        self.explore()
        return

    # Iterate through receptacles to find a suitable one
    for receptacle_entity in receptacle_entities:
        if not receptacle_entity.isopen and receptacle_entity.openable:
            # Open the receptacle if it's closed and can be opened
            self.open_receptacle(receptacle_entity.name)
            return
        elif receptacle_entity.isopen and object_entity.in_on == receptacle_entity.name:
            # If receptacle is open and object is already in it, do nothing
            return
        elif not object_entity.in_on:
            # If object is not in any receptacle, take it
            self.take(object_entity.name, object_entity.in_on)
            return
        elif not receptacle_entity.ison:
            # If receptacle is not holding anything, put the object in it
            self.put(object_entity.name, receptacle_entity.name)
            return
```

## I.2 PLANNING PROGRAMS IN NETHACK

### I.2.1 UPGRADE EXP LEVEL TO 3

```python
############### program 1 ###############
class NethackTemplate(SimpleSkillSelector):

    def init(self, agent, dict_obs: dict):
        """
            Initialize variables for this specific skill selector...
        """
        self.monster_list = []

    def update_init(self, agent, dict_obs: dict) -> dict:
        """
            Update variables for this specific skill selector...
        """
        self.monster_list = []
        for monster in dict_obs.get("Close Monsters", []) + dict_obs.get("Distant Monsters", []):
            if "tame " not in monster["item_name"] and monster["item_name"] not in ["shopkeeper", "gas
            ↪   spore"]:
                self.monster_list.append(monster)

    def core_function(self, agent, dict_obs: dict) -> dict:
        """
            Functionality of this function: Upgrade experience level to 3 in the 1st depth.
        """
        current_level = int(dict_obs["Stats"]["level"])
        thoughts_obs_prefix = f"Current level is {current_level}. Trying to reach level 3."

        if current_level >= 3:
```

```python
                observations = thoughts_obs_prefix + " Accomplished the task."
                reasoning = "Reached level 3."
                speak = "I have reached level 3. Task accomplished."
                skill = {
                    "name": "finish_task"
                }
        else:
            if not self.monster_list:
                observations = thoughts_obs_prefix + " No eligible monsters found."
                reasoning = "No eligible monsters found. Need to explore further."
                speak = "No eligible monsters found. Exploring further."
                skill = {
                    "name": "explore_level"
                }
            else:
                for monster in self.monster_list:
                    x, y = monster["x"], monster["y"]
                    if agent.get_path_to(x, y):
                        observations = thoughts_obs_prefix + f" Found a monster {monster['item_name']} to
                        ↪    attack."
                        reasoning = f"Found a monster {monster['item_name']} at ({x}, {y})."
                        speak = f"Attacking the monster {monster['item_name']} at ({x}, {y})."
                        skill = {
                            "name": "melee_attack",
                            "x": x,
                            "y": y
                        }
                        break
                else:
                    observations = thoughts_obs_prefix + " No path to any monster."
                    reasoning = "No valid path to any monster. Need to explore further."
                    speak = "No path to any monster. Exploring further."
                    skill = {
                        "name": "explore_level"
                    }

        json_dict = {
            "thoughts": {
                "observations": observations,
                "reasoning": reasoning,
                "speak": speak
            },
            "skill": skill
        }

        return json_dict

############### program 2 ###############
class NethackTemplate(SimpleSkillSelector):

    def init(self, agent, dict_obs: dict):
        # variables for this specific skill selector...
        self.target_level = 3
        self.corpse_locations = []   # Save locations of corpses to eat them later if needed
        self.tame_monsters = ["tame ", "shopkeeper", "gas spore"]

    def update_init(self, agent, dict_obs: dict) -> dict:
        for room in [dict_obs.get("Rooms", {"current_room": {"content_description":
        ↪    []}})["current_room"]] + \
                    dict_obs.get("Rooms", {"other_rooms": []})["other_rooms"]:
            for item in room["content_description"]:
                if "corpse" in item["item_name"] and (item["x"], item["y"]) not in self.corpse_locations:
                    self.corpse_locations.append((item["x"], item["y"]))

    def core_function(self, agent, dict_obs: dict) -> SkillSelection:
        """
        Upgrade your experience level to 3 in the 1st depth
        """

        stats = dict_obs["Stats"]
        current_level = stats["level"]
        thoughts_obs_prefix = f"I'm currently at experience level {current_level}."

        if current_level >= self.target_level:
            observations = thoughts_obs_prefix + f" I've reached the target experience level
            ↪    {self.target_level}."
            reasoning = "I need to finish the task."
            speak = f"I've reached the target experience level {self.target_level}. Task accomplished."

            skill = {
                "name": "finish_task"
```

```python
            }
        else:
            # Check for nearby monsters to attack
            for category in ["Close Monsters", "Distant Monsters"]:
                for monster in dict_obs.get(category, []):
                    if "tame " not in monster["item_name"] and monster["item_name"] not in
                    ↪    self.tame_monsters:
                        x, y = monster["x"], monster["y"]
                        distance = agent.get_distance_to(x, y)
                        if agent.get_path_to(x, y):
                            observations = thoughts_obs_prefix + f" I found an attackable monster:
                            ↪    {monster['item_name']} at ({x}, {y})."
                            if distance > 1:
                                reasoning = f"The monster is {distance} steps away, moving closer to
                                ↪    attack."
                                speak = f"Moving closer to attack the {monster['item_name']}."
                                skill = {
                                    "name": "move_to",
                                    "x": x,
                                    "y": y
                                }
                            else:
                                reasoning = f"The monster is within reach, attacking it now."
                                speak = f"Attacking the {monster['item_name']}."
                                skill = {
                                    "name": "melee_attack",
                                    "x": x,
                                    "y": y
                                }
                            return {
                                "thoughts": {
                                    "observations": observations,
                                    "reasoning": reasoning,
                                    "speak": speak
                                },
                                "skill": skill
                            }

        # If no enemies found, check food status
        if stats["hunger"] in ["weak", "fainting", "starved"]:
            # Check inventory for food
            for key, item in dict_obs.get("Inventory", {}).items():
                if item["item_name"] in food_names:
                    observations = thoughts_obs_prefix + " Found food in inventory."
                    reasoning = "I'm hungry and need to eat."
                    speak = "Eating some food from the inventory."
                    skill = {
                        "name": "eat",
                        "item_letter": key
                    }
                    return {
                        "thoughts": {
                            "observations": observations,
                            "reasoning": reasoning,
                            "speak": speak
                        },
                        "skill": skill
                    }

            # If no food in inventory, look for corpses to eat
            if self.corpse_locations:
                x, y = self.corpse_locations.pop(0)
                if agent.get_path_to(x, y):
                    observations = thoughts_obs_prefix + f" Found a corpse at ({x}, {y}) to eat."
                    if agent.get_distance_to(x, y) > 1:
                        reasoning = "The corpse is far away, moving closer."
                        speak = "Moving closer to the corpse to eat it."
                        skill = {
                            "name": "move_to",
                            "x": x,
                            "y": y
                        }
                    else:
                        reasoning = "The corpse is nearby, eating it now."
                        speak = "Eating the corpse from the ground."
                        skill = {
                            "name": "eat_from_ground",
                            "x": x,
                            "y": y
                        }
                    return {
```

```python
                    "thoughts": {
                        "observations": observations,
                        "reasoning": reasoning,
                        "speak": speak
                    },
                    "skill": skill
                }

        # If there's nothing to attack or eat, explore the level
        observations = thoughts_obs_prefix + " No enemies or food found nearby."
        reasoning = "I need to explore the level further."
        speak = "Exploring the level to find monsters or food."
        skill = {
            "name": "explore_level"
        }

        return {
            "thoughts": {
                "observations": observations,
                "reasoning": reasoning,
                "speak": speak
            },
            "skill": skill
        }

############### program 3 ###############
class NethackTemplate(SimpleSkillSelector):

    def init(self, agent, dict_obs: dict):
        """
            Initialize variables for this specific skill selector...
        """
        self.monsters_to_avoid = ["shopkeeper", "gas spore"]
        self.food_items = set()
        self.last_move = None

    def update_init(self, agent, dict_obs: dict) -> dict:
        """
            Update variables for this specific skill selector...
        """
        for room in [dict_obs.get("Rooms", {"current_room": {"content_description":
        ↪    []}})["current_room"]] + \
                    dict_obs.get("Rooms", {"other_rooms": []})["other_rooms"]:

            for item in room["content_description"]:
                if item["item_name"] in food_names:
                    self.food_items.add((item["x"], item["y"]))

    def core_function(self, agent, dict_obs: dict) -> SkillSelection:
        """
            Functionality of this function: Upgrade experience level to 3 in the 1st depth.
        """
        # Check for current experience level
        current_exp = dict_obs["Stats"]["exp"]
        if current_exp >= 60:  # Assume level 3 is reached at 60 experience points
            return {
                "thoughts": {
                    "observations": f"Current experience is {current_exp}.",
                    "reasoning": "Experience level 3 has been achieved.",
                    "speak": "I have reached experience level 3. Task accomplished."
                },
                "skill": {
                    "name": "finish_task"
                }
            }

        # Find and attack monsters
        for monster in dict_obs.get("Close Monsters", []):
            if "tame " not in monster["item_name"] and monster["item_name"] not in
            ↪    self.monsters_to_avoid:
                x, y = monster["x"], monster["y"]
                if agent.get_path_to(x, y):
                    return {
                        "thoughts": {
                            "observations": f"Found monster {monster['item_name']} at ({x}, {y}).",
                            "reasoning": f"The monster {monster['item_name']} is in attacking range.",
                            "speak": f"Attacking {monster['item_name']}."
                        },
                        "skill": {
                            "name": "melee_attack",
                            "x": x,
```

```python
                                "y": y
                            }
                    }

            # Handle hunger
            hunger_status = dict_obs["Stats"]["hunger"]
            if hunger_status in ["hungry", "weak", "fainting", "starved"]:
                if dict_obs["Inventory"]:
                    for key, item in dict_obs["Inventory"].items():
                        if item["item_name"] in food_names:
                            return {
                                "thoughts": {
                                    "observations": f"Hungry status detected. Found {item['item_name']} in
                                    ↪   inventory.",
                                    "reasoning": f"Need to eat {item['item_name']} to avoid starvation.",
                                    "speak": f"Eating {item['item_name']} to survive."
                                },
                                "skill": {
                                    "name": "eat",
                                    "item_letter": key
                                }
                            }
                for x, y in self.food_items:
                    if agent.get_path_to(x, y):
                        return {
                            "thoughts": {
                                "observations": f"Hungry status detected. Found food on the ground at ({x},
                                ↪   {y}).",
                                "reasoning": "Need to move to the food to eat it.",
                                "speak": "Moving to food to avoid starvation."
                            },
                            "skill": {
                                "name": "move_to",
                                "x": x,
                                "y": y
                            }
                        }
                    else:
                        self.food_items.discard((x, y))

            # Explore the level to find monsters or items
            return {
                "thoughts": {
                    "observations": "No immediate actions to take.",
                    "reasoning": "Exploring the level to find monsters to gain experience.",
                    "speak": "Exploring the level."
                },
                "skill": {
                    "name": "explore_level"
                }
            }
```

## I.3 PLANNING PROGRAMS IN SC2

### I.3.1 HARD

```python
############### program 1 ###############
import math
def planner(obs, action_space, task):
    '''
    Parameters:
        obs is a dict with the following format:
            {
                "Resource": {
                    "supply_cap": 15,
                    "supply_left": 3,
                    "gas": 0
                },
                "Building": {
                    "COMMANDCENTER": 1,
                    "BARRACKS": 0,
                    "SUPPLYDEPOT": 0,
                    "REFINERY": 0,
                    // ...
                },
                "Unit": {
                    "SCV": 12,
                    "SIEGETANK": 0,
```

```python
                    // ...
                }
            }
            with the specified number of each resource/building/unit in the current game state

        action_space is a list of strings including all the available actions

        task is a unit dict:
            {
                "SCV": "num_1",
                "SIEGETANK": "num_2",
                "VIKINGFIGHTER": "num_3",
                "MEDIVAC": "num_4",
                "GHOST": "num_5",
            }
            with the goal of training the specified quantities of the corresponding type of units in
↪   the game.
    '''
    plan_build = []
    plan_unit = []

    # infer the tech_tree from the unit of the task
    tech_tree = {
        "SCV": {
            "base_building": "COMMANDCENTER",
            "pre_dependency": {},
        },
        "SIEGETANK": {
            "base_building": "FACTORYTECHLAB",
            "pre_dependency": {
                1: "FACTORY",
                2: "BARRACKS",
            },
        },
        "VIKINGFIGHTER": {
            "base_building": "STARPORT",
            "pre_dependency": {
                1: "STARPORTTECHLAB",
                2: "STARPORT",
            },
        },
        "MEDIVAC": {
            "base_building": "STARPORT",
            "pre_dependency": {
                1: "STARPORT",
            },
        },
        "GHOST": {
            "base_building": "BARRACKSTECHLAB",
            "pre_dependency": {
                1: "BARRACKS",
                2: "GHOSTACADEMY",
            },
        },
    }
    # obtain the base_building for the technology
    base_buildings = {k: v["base_building"] for k, v in tech_tree.items()}

    '''
        when supply_left is less than 8, increasing supply_cap (BUILD SUPPLYDEPOT) is necessary.
    '''
    if obs["Resource"]["supply_left"] < 8:
        if "BUILD SUPPLYDEPOT" in action_space:
            plan_build.append("BUILD SUPPLYDEPOT")

    '''
        gas is important, check if there is a need to collecting gas (BUILD REFINERY).
    '''
    if "BUILD REFINERY" in action_space and obs["Resource"]["gas"] == 0:
        plan_build.append("BUILD REFINERY")

    '''
        Check the 'unit' that still need to be trained in the current game state, and add f'TRAIN {unit}'
↪   to the plan_unit for each unit in units. You need to ensure that f'TRAIN {unit}' is in the
↪   action_space.
    '''
    unit_still_needed_num = {unit: max(0, target_num - obs["Unit"][unit]) for unit, target_num in
↪       task.items()}
    for unit, target_num in unit_still_needed_num.items():
        if f"TRAIN {unit}" in action_space and target_num > 0:
            plan_unit.append(f"TRAIN {unit}")
```

```python
    '''
        calculate the number still needed for each base_building in the task
    '''
    scale_of_scv_per_base_building = 16
    scale_of_otherunit_per_base_building = 8
    base_building_needed_num = {building: 0 for _, building in base_buildings.items()}
    for unit, target_num in task.items():
        if unit == "SCV":
            base_building_needed_num[base_buildings[unit]] += math.ceil(task[unit] /
            ↪  scale_of_scv_per_base_building)
        else:
            base_building_needed_num[base_buildings[unit]] += math.ceil(task[unit] /
            ↪  scale_of_otherunit_per_base_building)

    '''
        Based on the tech_tree, analyze which 'building' are still needed for each unit in the task at the
↪  current game state. Then add f'BUILD {building}' to the plan_build for each building in required
↪  buildings. You need to ensure that f'BUILD {building}' is in the action_space.
    '''
    for unit, tech in tech_tree.items():
        pre_dependency = tech.get("pre_dependency")
        base_building = tech.get("base_building")
        # first check pre_dependency, as only when the pre_dependency is met can the base_building be
        ↪  constructed.
        if pre_dependency:
            pre_dependency = dict(sorted(pre_dependency.items(), key=lambda x: x[0]))
            for priority, building in pre_dependency.items():
                # only need 1 for each building in pre_dependency
                if f"BUILD {building}" in action_space and obs["Building"][building] == 0:
                    plan_build.append(f"BUILD {building}")

        # then check the base_building
        if f"BUILD {base_building}" in action_space and obs["Building"][base_building] <
        ↪  base_building_needed_num[base_building]:
            plan_build.append(f"BUILD {base_building}")

    # mix the plan_build and plan_unit alternately to get the plan
    plan = []
    while plan_build or plan_unit:
        if plan_build:
            plan.append(plan_build.pop(0))
        if plan_unit:
            plan.append(plan_unit.pop(0))

    # return the first 5 actions as a plan
    return plan[:5]

############## program 2 ##############
import math
def planner(obs, action_space, task):
    '''
    Parameters:
        obs is a dict with the following format:
            {
                "Resource": {
                    "supply_cap": 15,
                    "supply_left": 3,
                    "gas": 0
                },
                "Building": {
                    "COMMANDCENTER": 1,
                    "BARRACKS": 0,
                    "SUPPLYDEPOT": 0,
                    "REFINERY": 0,
                    // ...
                },
                "Unit": {
                    "SCV": 12,
                    "MARINE": 0,
                    // ...
                }
            }
            with the specified number of each resource/building/unit in the current game state

        action_space is a list of strings including all the available actions

        task is a unit dict:
            {
                "SCV": "num_1",
                "SIEGETANK": "num_2",
```

```python
                "VIKINGFIGHTER": "num_3",
                "MEDIVAC": "num_4",
                "GHOST": "num_5",
            }
            with the goal of training the specified quantities of the corresponding type of units in
↪    the game.
    '''
    plan_build = []
    plan_unit = []

    tech_tree = {
        "SCV": {
            "base_building": "COMMANDCENTER",
            "pre_dependency": {},
        },
        "SIEGETANK": {
            "base_building": "FACTORYTECHLAB",
            "pre_dependency": {
                1: "FACTORY",
                2: "ARMORY",
            },
        },
        "VIKINGFIGHTER": {
            "base_building": "STARPORTTECHLAB",
            "pre_dependency": {
                1: "STARPORT",
            },
        },
        "MEDIVAC": {
            "base_building": "STARPORT",
            "pre_dependency": {
                1: "STARPORT",
            },
        },
        "GHOST": {
            "base_building": "GHOSTACADEMY",
            "pre_dependency": {
                1: "BARRACKSTECHLAB",
                2: "BARRACKS",
            },
        }
    }
    base_buildings = {k: v["base_building"] for k, v in tech_tree.items()}

    if obs["Resource"]["supply_left"] < 8:
        if "BUILD SUPPLYDEPOT" in action_space:
            plan_build.append("BUILD SUPPLYDEPOT")

    if "BUILD REFINERY" in action_space and obs["Resource"]["gas"] == 0:
        plan_build.append("BUILD REFINERY")

    unit_still_needed_num = {unit: max(0, target_num - obs["Unit"][unit]) for unit, target_num in
↪    task.items()}
    for unit, target_num in unit_still_needed_num.items():
        if f"TRAIN {unit}" in action_space and target_num > 0:
            plan_unit.append(f"TRAIN {unit}")

    scale_of_scv_per_base_building = 16
    scale_of_otherunit_per_base_building = 8
    base_building_needed_num = {building: 0 for _, building in base_buildings.items()}
    for unit, target_num in task.items():
        if unit == "SCV":
            base_building_needed_num[base_buildings[unit]] += math.ceil(task[unit] /
            ↪    scale_of_scv_per_base_building)
        else:
            base_building_needed_num[base_buildings[unit]] += math.ceil(task[unit] /
            ↪    scale_of_otherunit_per_base_building)

    for unit, tech in tech_tree.items():
        pre_dependency = tech.get("pre_dependency")
        base_building = tech.get("base_building")
        if pre_dependency:
            pre_dependency = dict(sorted(pre_dependency.items(), key=lambda x: x[0]))
            for priority, building in pre_dependency.items():
                if f"BUILD {building}" in action_space and obs["Building"][building] == 0:
                    plan_build.append(f"BUILD {building}")

        if f"BUILD {base_building}" in action_space and obs["Building"][base_building] <
        ↪    base_building_needed_num[base_building]:
            plan_build.append(f"BUILD {base_building}")
```

```python
    plan = []
    while plan_build or plan_unit:
        if plan_build:
            plan.append(plan_build.pop(0))
        if plan_unit:
            plan.append(plan_unit.pop(0))

    return plan[:5]

############### program 3 ###############
import math
def planner(obs, action_space, task):
    '''
        Parameters:
            obs is a dict with the following format:
                {
                    "Resource": {
                        "supply_cap": 15,
                        "supply_left": 3,
                        "gas": 0
                    },
                    "Building": {
                        "COMMANDCENTER": 1,
                        "BARRACKS": 0,
                        "SUPPLYDEPOT": 0,
                        "REFINERY": 0,
                        // ...
                    },
                    "Unit": {
                        "SCV": 12,
                        "MARINE": 0,
                        // ...
                    }
                }
                with the specified number of each resource/building/unit in the current game state

            action_space is a list of strings including all the available actions

            task is a unit dict:
                {
                    "SCV": "num_1",
                    "SIEGETANK": "num_2",
                    "VIKINGFIGHTER": "num_3",
                    "MEDIVAC": "num_4",
                    "GHOST": "num_5",
                }
                with the goal of training the specified quantities of the corresponding type of units in
the game.
    '''
    plan_build = []
    plan_unit = []

    # infer the tech_tree from the unit of the task
    tech_tree = {
        "SCV": {
            "base_building": "COMMANDCENTER",
            "pre_dependency": {},
        },
        "SIEGETANK": {
            "base_building": "FACTORYTECHLAB",
            "pre_dependency": {
                1: "FACTORY",
                2: "ARMORY",
            },
        },
        "VIKINGFIGHTER": {
            "base_building": "STARPORTTECHLAB",
            "pre_dependency": {
                1: "STARPORT",
            },
        },
        "MEDIVAC": {
            "base_building": "STARPORT",
            "pre_dependency": {
                1: "SUPPLYDEPOT",
                2: "BARRACKS",
            },
        },
        "GHOST": {
            "base_building": "GHOSTACADEMY",
            "pre_dependency": {
```

```python
                1: "BARRACKSTECHLAB",
                2: "FACTORY",
            },
        }
    }
    # obtain the base_building for the technology
    base_buildings = {k: v["base_building"] for k, v in tech_tree.items()}

    '''
        when supply_left is less than 8, increasing supply_cap (BUILD SUPPLYDEPOT) is necessary.
    '''
    if obs["Resource"]["supply_left"] < 8:
        if "BUILD SUPPLYDEPOT" in action_space:
            plan_build.append("BUILD SUPPLYDEPOT")

    '''
        gas is important, check if there is a need to collecting gas (BUILD REFINERY).
    '''
    if "BUILD REFINERY" in action_space and obs["Resource"]["gas"] == 0:
        plan_build.append("BUILD REFINERY")

    '''
        Check the 'unit' that still need to be trained in the current game state, and add f'TRAIN {unit}'
↪   to the plan_unit for each unit in units. You need to ensure that f'TRAIN {unit}' is in the
↪   action_space.
    '''
    unit_still_needed_num = {unit: max(0, target_num - obs["Unit"][unit]) for unit, target_num in
↪   task.items()}
    for unit, target_num in unit_still_needed_num.items():
        if f"TRAIN {unit}" in action_space and target_num > 0:
            plan_unit.append(f"TRAIN {unit}")

    '''
        calculate the number still needed for each base_building in the task
    '''
    scale_of_scv_per_base_building = 16
    scale_of_otherunit_per_base_building = 8
    base_building_needed_num = {building: 0 for _, building in base_buildings.items()}
    for unit, target_num in task.items():
        if unit == "SCV":
            base_building_needed_num[base_buildings[unit]] += math.ceil(task[unit] /
↪           scale_of_scv_per_base_building)
        else:
            base_building_needed_num[base_buildings[unit]] += math.ceil(task[unit] /
↪           scale_of_otherunit_per_base_building)

    '''
        Based on the tech_tree, analyze which 'building' are still needed for each unit in the task at the
↪   current game state. Then add f'BUILD {building}' to the plan_build for each building in required
↪   buildings. You need to ensure that f'BUILD {building}' is in the action_space.
    '''
    for unit, tech in tech_tree.items():
        pre_dependency = tech.get("pre_dependency")
        base_building = tech.get("base_building")
        # first check pre_dependency, as only when the pre_dependency is met can the base_building be
↪       constructed.
        if pre_dependency:
            pre_dependency = dict(sorted(pre_dependency.items(), key=lambda x: x[0]))
            for priority, building in pre_dependency.items():
                # only need 1 for each building in pre_dependency
                if f"BUILD {building}" in action_space and obs["Building"][building] == 0:
                    plan_build.append(f"BUILD {building}")

        # then check the base_building
        if f"BUILD {base_building}" in action_space and obs["Building"][base_building] <
↪       base_building_needed_num[base_building]:
            plan_build.append(f"BUILD {base_building}")

    # mix the plan_build and plan_unit alternately to get the plan
    plan = []
    while plan_build or plan_unit:
        if plan_build:
            plan.append(plan_build.pop(0))
        if plan_unit:
            plan.append(plan_unit.pop(0))

    # return the first 5 actions as a plan
    return plan[:5]
```