# OpenReview forum: "Code Driven Planning with Domain-Adaptive Selector"
_ICLR.cc/2026/Conference — ICLR 2026 Poster_

### Official Review · Reviewer_R9TH · 2025-10-31

**Soundness:** 2
**Presentation:** 2
**Contribution:** 3
**Rating:** 6
**Confidence:** 3

**Summary:**

This work introduces CoPic, a framework to allow LLMs to generate planning-programs, which are used to generate and refine a set of candidate plans for the agent. These planning programs are generated by a LLM Planner, which are able to cheaply and efficiently generate plans. The selector selects the best plan from the plans generated by all of the programs. At runtime, this set of planning programs are used to generate candidate plans which the selector picks from to execute. During the learning phase, the history of the executions are used to refine the programs, as well as finetune the selector with RL. Results on 3 different environments show that CoPic exhibits better data efficiency and cost reduction while performing better than existing methods.

**Strengths:**

This work evaluates their method on several different and challenging benchmarks, and also against many different strong baselines, exhibiting strong benchmark results. Analysis on the Planning Quality as well as the Planning Efficiency in terms of cost is meticulous and well documented in order to show that their technique shows better performance while reducing costs. Ablations in Section 5.4, notably on the open-source LLMs as well as the program evolution are targeted and insightful.

**Weaknesses:**

One weakness would be the lack of ablations around the selector module itself. The selector, as detailed in 4.1.2, is trained to output a scoring of the plans as a probability distribution across each of the candidate plans. This small selector is the only module that is trained with gradient based methods and RL, which the other non-training based baselines such as AdaPlanner and Prospector do not use. There is no ablation on this module itself, except in Section 5.4.3, where they do an ablation against a random selector policy.

One obvious ablation and experiment would be to compare with using a prompted LLM-as-a-Judge to simply select the plans that are generated by the planning programs. This ablation would provide greater insight into how this RL finetuning affects the performance.

Similarly, the baselines also lack any LLM training-based methods, opting only for feedback-based LLM planning methods [line 337] and Non-LLM training-based methods [lines 339-340]. There is already training data used during the learning phase (Section 4.2.1), so why not just do finetuning instead, maybe with some filtered behavior cloning [1], or even generating plan data and finetuning a planner policy. [2]

From a presentation standpoint, Section 4 is somewhat difficult to read and understand. It is unclear what a "planning program" is, from a first read and how this kind of planning program actually generates plans to be executed. Presentation could be clearer on the structure of this kind of program as well as what the output looks like.

[1] Pan, Jiayi, et al. "Autonomous Evaluation and Refinement of Digital Agents." First Conference on Language Modeling.
[2] Erdogan, Lutfi Eren, et al. "Plan-and-Act: Improving Planning of Agents for Long-Horizon Tasks." Forty-second International Conference on Machine Learning.

**Questions:**

During the learning phase, in Section 4.2.1, is the LLM Planner itself learning from the experience of generating planning programs at all? For instance, when it gets feedback on the performance of the plans and it updates the plans in the environment, is there any feedback to the LLM Planner itself so that it can generate better planning programs in the future? If so, it would be great to see how the performance of the initial plans of the LLM Planner performs as it sees more and more tasks.

---

> ### Author Response · Authors · 2025-11-20
> **Rebuttal -- Part 1**
>
> Thanks for your detailed review. We are glad to discuss your concerns one by one. Any further discussion will be appreciated.
> Besides, we have revised our paper to address your concerns. We summarize the main changes as the following:
>
> (1) Add the result of CoPiC(LaJ)—that is, CoPiC taking LLM-as-a-Judge as the selector—in Table 1. (**Weakness 1**)
>
> (2) Add a new baseline, TWOSOME, in Table 1. (**Weakness 2**)
>
> (3) Add an example of planning program in Section 4.1.1 for aiding understanding. (**Weakness 3**)
>
> (4) Add an analysis of the impact of experiences from other type of tasks on CoPiC's initial performance in Appendix E.9. (**Question 1**)
>
>
> > Weakness 1. The methods in the baselines do not employ gradient-based training. Taking LLM-as-a-Judge as the selector for ablation.
>
>
> - In fact, **Prospector—one of the baselines—also incorporates a training phase**: it supervises an LLM-critic for trajectory selection via supervised fine-tuning (SFT) on expert demonstrations. By contrast, CoPiC performs training with reinforcement learning driven by autonomous exploration, eliminating the need for curated expert data. It can be seen from Table 1 in the paper that CoPiC surpasses Prospector by 24.53% in success rate while reducing token consumption by 88.70%, which demonstrates CoPiC's superiority in both performance and efficiency.
>
> - To address your concern, we instantiated the CoPiC selector as an LLM-as-a-Judge and compared it with our RL-finetuned version. The results are shown below, with table entries formatted as **Success Rate (Cost)**. Compared with LLM-as-a-Judge (GPT4.1-as-a-Judge), the RL-finetuned variant improves the success rate by 19.88% and reduces token consumption by 27.14%. This gain arises because **the RL-finetuned selector acquires domain-specific knowledge during interaction, yielding more accurate plan evaluations than the prior-free LLM-as-a-Judge approach.**
>
>     | Method | Pick | Examine | Clean | Heat | Cool | Pick Two |
>     |---------|---------|---------| --------- | --------- | --------- | --------- |
>     | CoPiC (RL-Fintuned Selector) | 100.00 (0.05M) | 100.00 (0.04M) | 100.00 (0.33M) | 100.00 (0.26M) | 100.00 (0.28M) | 95.29 (0.06M) |
>     | CoPiC (GPT4.1-as-a-Judge) | 84.17 (0.08M) | 78.89 (0.09M) | 78.71 (0.42M) | 80.87 (0.33M) | 78.10 (0.37M) | 75.29 (0.11M) |
>
> > Weakness 2. The baselines lack any LLM training-based methods. Why do not directly training a planner policy using data?
>
>
> - Actually, **Prospector—one of the baselines—also incorporates a training phase**: it supervises an LLM-critic for trajectory selection via supervised fine-tuning (SFT) on expert demonstrations. By contrast, CoPiC performs training with reinforcement learning driven by autonomous exploration, eliminating the need for curated expert data. It can be seen from Table 1 in the paper that CoPiC surpasses Prospector by 24.53% in success rate while reducing token consumption by 88.70%, which demonstrates CoPiC's superiority in both performance and efficiency.
>
> - To address your concern, we include TWOSOME [1] as an additional baseline. **TWOSOME leverages an LLM as its backbone, training it as a PPO policy to output task plans directly, with fine-tuning data derived from policy-environment interactions**. The comparative results are reported below, with table entries formatted as **Success Rate (Cost)**. It can be seen that CoPiC surpasses TWOSOME by 22.29% in success rate while reducing token consumption by 78.16%.
>
>
>     | Method | Pick | Examine | Clean | Heat | Cool | Pick Two |
>     |---------|---------|---------| --------- | --------- | --------- | --------- |
>     | CoPiC | 100.00 (0.05M) | 100.00 (0.04M) | 100.00 (0.33M) | 100.00 (0.26M) | 100.00 (0.28M) | 95.29 (0.06M) |
>     | TWOSOME | 71.67 (0.32M) | 74.44 (0.19M) | 76.78 (1.24M) | 76.52 (1.08M) | 80.95 (1.12M) | 81.18 (0.72M) |
>
>
> [1] Tan et al. True Knowledge Comes from Practice: Aligning LLMs with Embodied Environments via Reinforcement Learning
>
>
>
> > Weakness 3. Section 4 is somewhat difficult to read and understand.
>
> We appreciate your pointing out the unclear expressions in the manuscript. We clarify that a planning program is a program capable of interacting with the environment in a closed-loop paradigm. **The input to this program is the current observation from the environment, and the output plan is an action sequence generated based on that observation. This plan is then used to interact with the environment to advance task completion.** To aid understanding, we have added a simple example of planning program in Section 4.1.1.

---

> ### Author Response · Authors · 2025-11-20
> **Rebuttal -- Part 2**
>
> > Question 1. During the learning phase, in Section 4.2.1, is the LLM Planner itself learning from the experience of generating planning programs at all? If so, it would be great to see how the performance of the initial plans of the LLM Planner performs as it sees more and more tasks.
>
> Dear Reviewer. Given that CoPiC constructs planning programs and selectors for a category of tasks, we are somewhat unclear about the specific aspect you are inquiring about. Specifically, we are uncertain whether you are asking **(1)** if the LLM continuously refines the planning programs based on interaction experiences while completing a category of tasks, or **(2)** if the LLM utilizes interaction experiences from previously completed task categories to initialize planning programs when addressing a new category of tasks.
>
> - If your question pertains to (1),
>
>     - Yes, **the LLM improves the planning programs based on interaction experiences with the environment**, a process referred to as "Programs Evolution". The LLM evolves the planning programs every 20 episodes, meaning that after every 20 episodes, the LLM refines the current planning programs based on the accumulated interaction experiences. You may refer to Algorithm 1 in Appendix G for a clear understanding of the CoPiC workflow.
>
>     - In fact, Figure 3 in Section 5 of this paper already illustrates the gradual improvement of CoPiC's performance as trails progress. **The point on the horizontal axis where Trails = 0 corresponds to the performance of the initial plans of planning programs without any evolution.**
>
> - If your question pertains to (2),
>     - In the current results, the LLM does not utilize interaction experiences from previously completed task categories to initialize planning programs when addressing a new category of tasks. However, to address your question, we **compared the performance of CoPiC's planning programs initialized with and without interaction experiences** from the Pick task on the Clean, Heat, and Cool tasks in ALFWorld. The results are presented below. It can be observed that with interaction experiences from previously completed tasks incorporated as part of the prompt, the success rate of CoPiC's initially generated planning programs increased by 11.63%. This indicates that **LLM can enhance its capability to generate initial plans by leveraging existing experiences**.
>
>         | Method | Clean | Heat | Cool |
>         |---------|---------|---------| --------- |
>         | CoPiC w Pick Exp| 43.87 | 53.91 | 14.29 |
>         | CoPiC w/o Pick Exp | 38.06 | 39.13 | 0.00 |

---

> > ### Comment · Reviewer_R9TH · 2025-11-26
> >
> > I thank the authors for their thorough responses.
> >
> > Based on the additional experiments given, I have raised my soundness score and my confidence score.
> >
> > Regarding the writing, I still think that that section is a bit unclear on what exactly a planning program is. Could you maybe put a toy example of a planning program, something simple and easy to understand, so that readers can get a better grasp of what a planning program looks like? It can be a simplified version of some of the tasks in your experiments, or a completely different toy example. With this, I would be willing to raise the presentation score.
> >
> > I think that the results shown have shown that the method has merits and improvements over prior work such as Prospector, Adaplanner. However, I would like some more indepth analysis of the method in order to have greater insights.
> >
> > 1. I appreciate the scaling experiments in Appendix E.6 which show how the scaling of plans affects performance. However, in this case, n=3 was chosen because it had already saturated performance at 100%. Could this analysis be done on a more difficult task where it is not saturated at n=3, to see if the performance improves or degrades? Basically, I would like to see some kind of inference time scaling experiment in this direction.
> >
> > 2. Is there some insight to be gained on how often the trained adapter changes plans? At each step, the planning programs generate some plans and the domain specific adapter can choose to follow the plan of the planning program from the prior step, or it can pick the output of another program. I would like to see some kind of analysis on the frequency at which the adaptor swaps between plans. Does it follow a plan for a couple of steps and then switch, or is it basically switching at every step?
> >
> > If either of the above can be addressed, I would be willing to raise the contribution score and the rating score.

---

> > > ### Author Response · Authors · 2025-11-28
> > >
> > > Thank you again for your thoughtful review. We have revised our paper to address your concerns. We summarize the main changes as the following:
> > >
> > > (1) Replace the example of planning program in **Section 4.1.1** with a simple and complete example.
> > >
> > > (2) Add an analysis of how the number of planning programs affects performance on the SuperHard task in **Appendix E.11**.
> > >
> > > (3) Add an analysis of the selector’s frequency in switching planning programs in **Appendix E.12**.
> > >
> > > > Example of Planning Program
> > >
> > > We have replaced the example in Section 4.1.1 of the paper with the following simple and complete planning program example for 'Building SCV' in StarCraft II. This example intuitively shows to readers with 'what a planning program looks like'.
> > >
> > > ```python
> > > def planner(obs, action_space, task):
> > >     '''
> > >     obs is a dict with the specified number of each resource/building/unit at the current game state;
> > >     action_space is a list of strings including all the available actions;
> > >     task is a unit dict: {"SCV": num_1}, with the goal of building num_1 SCVs.
> > >     '''
> > >     plan_build = []
> > >     plan_unit = []
> > >
> > >     # infer the tech_tree for SCV
> > >     tech_tree = {"SCV": {"base_building": "COMMANDCENTER", "pre_dependency": []}}
> > >
> > >     # when supply_left is less than 8, increasing supply_cap (BUILD SUPPLYDEPOT) is necessary.
> > >     if obs["Resource"]["supply_left"] < 8:
> > >         if "BUILD SUPPLYDEPOT" in action_space:
> > >             plan_build.append("BUILD SUPPLYDEPOT")
> > >     # gas is important, check if there is a need to collecting gas (BUILD REFINERY).
> > >     if obs["Resource"]["gas"] == 0 and "BUILD REFINERY" in action_space:
> > >         plan_build.append("BUILD REFINERY")
> > >
> > >     # Check the number of SCV that still need to be trained at the current game state.
> > >     scv_still_needed_num = task["SCV"] - obs["Unit"]["SCV"]
> > >     if scv_still_needed_num > 0 and f"TRAIN SCV" in action_space:
> > >         plan_unit.append(f"TRAIN SCV")
> > >     # Analyze which 'building' are still needed for buildinig SCV at the current game state.
> > >     scv_base_building = tech_tree["SCV"]["base_building"]
> > >     if obs["Building"][scv_base_building] < 1 and f"BUILD {scv_base_building}" in action_space:
> > >         plan_build.append(f"BUILD {scv_base_building}")
> > >     plan = []
> > >     while plan_build or plan_unit:
> > >         if plan_build:
> > >             plan.append(plan_build.pop(0))
> > >         if plan_unit:
> > >             plan.append(plan_unit.pop(0))
> > >
> > >     return plan
> > > ```
> > >
> > >
> > > > Analysis of $n$ on a more difficult task
> > >
> > > We further constructed the **SuperHard** task on the StarCraft II Unit Building environment, which requires building all 17 Terran unit types. The results for varying values of  $n$ on this task are shown below. **It can be seen that on the SuperHard task, the number of required planning programs has also increased to 5.** The performance of the $n=6$ slightly drops compared with $n=5$, because a larger n raises the difficulty for the selector to pick the plan most aligned with the long-term reward—a trade-off between the number of programs and performance. Overall, **the greater variety of unit types implies a more complex tech tree, and to fully construct this complex tech tree, more planning programs are needed to work in coordination.** And CoPiC's diverse planning programs, combined with the domain-adaptive selector, enable it to scale and tackle such complex tasks.
> > >
> > > | n | SR | Cost |
> > > |---------|---------|---------|
> > > | 2 | 32.00% | 1.71M |
> > > | 3 | 44.00% | 2.14M |
> > > | 4 | 72.00% | 3.92M |
> > > | 5 | 96.00% | 2.42M |
> > > | 6 | 92.00% | 3.88M |
> > >
> > > > Analysis of the selector's frequency in switching planning programs
> > >
> > > The table below shows the intervals at which the Domain-Adaptive Selector switches planning programs to generate plans in ALFWorld. **The results indicate that for simpler tasks, such as Pick, the switching intervals are longer (i.e., lower frequency), while for more complex tasks, such as Examine and Clean, the switching intervals are shorter (i.e., higher frequency).** This finding suggests that **as task complexity increases, the number of necessary planning programs correspondingly rises**. Consequently, **the selector needs to switch plans more frequently in these more challenging tasks to adapt to their rising complexity.** This also shows that generating a single planning program capable of completing the task is difficult, whereas CoPiC produces diverse planning programs and, with the domain-adaptive selector, accomplishes the task at a lower cost.
> > >
> > > | Task | Pick | Examine | Clean| Heat | Cool | PickTwo |
> > > |---------|---------|---------|---------|---------|---------|---------|
> > > | Switch Interval | 2.78 | 1.52 | 1.34 | 1.68 | 1.76 | 2.32 |
> > >
> > >
> > > We hope the responses above can address your concerns. As you kindly noted, we would appreciate it if you could update your scores accordingly. Further discussion is always welcome.

---

### Official Review · Reviewer_4nz9 · 2025-10-31

**Soundness:** 2
**Presentation:** 3
**Contribution:** 2
**Rating:** 4
**Confidence:** 2

**Summary:**

This paper introduces CoPiC, a planning framework that leverages LLMs and RL. Unlike existing methods that rely on frequent, costly LLM queries and short-term feedback, CoPiC generates multiple high-level plans (using LLMs) and employs a domain-adaptive selector to choose the plan most aligned with long-term rewards. The selector is fine-tuned via reinforcement learning.
Solution was tested in different problems and showed improvements over baselines on both cost and SR.

**Strengths:**

Reduces LLM query costs while improving planning quality.
Demonstrates generalization and domain adaptation across diffferent environments.

**Weaknesses:**

Solutions are highligh dependent on the quality of the generated plans.
Selector’s effectiveness depends on RL fine-tuning.

**Questions:**

- PPDL approaches that use LLM alonside additional solvers have the potential to also be more cost effective and achieve good performance. Have you considered using one of these solutions as baseline?
- How sensitive is CoPiC’s performance to the diversity and quality of the generated plans?
- How does the number of generated plans affect performance (and cost)?

---

> ### Author Response · Authors · 2025-11-20
> **Rebuttal -- Part 1**
>
> Thanks for your detailed review. We are glad to discuss your concerns one by one. Any further discussion will be appreciated.
> Besides, we have revised our paper to address your concerns. We summarize the main changes as the following:
>
> (1) Add a new baseline, the PDDL-based method LLM-DP, in Table 1. (**Question 1**)
>
> (2) Add an analysis of the impact of temperature variations on CoPiC's performance in Appendix E.8. (**Qeustion 2 && 3**)
>
>
> > Weakness. Solutions are highligh dependent on the quality of the generated plans. Selector’s effectiveness depends on RL fine-tuning.
>
> Dear Reviewer. We apologize for any confusion, but we were unable to identify the specific weakness you are referring to. Below is our response. If our answer does not address your concern, we welcome further discussion to clarify the matter.
>
> The two points you mentioned **are precisely the core innovations of CoPiC, rather than weaknesses**.
>
> - Compared to the baselines' planning optimization based on context prompts, **the iterative optimization design of CoPiC's planning programs and selector** can further enhance the quality of planning. CoPiC generates candidate plans through planning programs, and then the selector identifies the one most aligned with long-term rewards for execution, thereby producing domain-specific interaction experiences. These experiences are used to instruct the LLM to evolve the planning programs, thereby enhancing the quality of the candidate plans. Additionally, the experiences are utilized in reinforcement learning (RL) to train the selector, increasing the accuracy of its evaluations on candidate plans. **This iterative optimization of planning programs and the selector leads to the progressive enhancement of CoPiC's performance**, thereby demonstrating superior and more efficient planning optimization compared to the baselines that rely solely on context prompts.
>
> - RL fine-tuning enables the selector to **acquire domain-specific knowledge**, thereby enhancing the accuracy of its evaluations of candidate plans. The data used for RL fine-tuning consists of interaction experiences with the environment. Based on the reward signals from the environment, the selector can **progressively learn to accurately assess the alignment of plans with long-term rewards**.
>
> > Question 1. PDDL approaches that use LLM alonside additional solvers have the potential to also be more cost effective and achieve good performance. Have you considered using one of these solutions as baseline?
>
>
> PDDL-based methods rely on additional domain files, problem files, and solvers to search the problem space for solutions, **whereas the definition of domain and problem files requires substantial manual effort**. In contrast, **CoPiC eliminates the need for both domain and problem files, thereby offering greater generality**.
>
> However, to address your question, we take a PDDL-based approach, specifically LLM-DP [1], as a baseline. The results are as follows, with table entries formatted as **Success Rate (Cost)**. It demonstrates that, compared to LLM-DP, CoPiC eliminates the need for additional human effort in defining domain and problem files, while achieving a success rate higher by 8.56%, and reduces token consumption by 78.11%.
>
> | Method | Pick | Examine | Clean | Heat | Cool | Pick Two |
> |---------|---------|---------| --------- | --------- | --------- | --------- |
> | CoPiC | 100.00 (0.05M) | 100.00 (0.04M) | 100.00 (0.33M) | 100.00 (0.26M) | 100.00 (0.28M) | 95.29 (0.06M) |
> | LLM-DP | 91.67 (0.61M) | 95.56 (0.21M) | 92.90 (0.78M) | 92.17 (0.81M) | 90.48 (1.17M) | 81.18 (1.08M) |
>
> [1] Dagan et al. Dynamic Planning with a LLM. NeurIPS 2024 Workshop LanGame

---

> ### Author Response · Authors · 2025-11-20
> **Rebuttal -- Part 2**
>
> > Question 2 && 3. How sensitive is CoPiC’s performance to the diversity and quality of the generated plans? How does the number of generated plans affect performance (and cost)?
>
> - **Diversity**. Diversity is determined by the temperature of the LLM. The following ablation studies, with table entries formatted as **Success Rate (Cost)**, on different temperatures demonstrate that temperature has a certain impact on CoPiC's performance and cost, but CoPiC is not highly sensitive to temperature variations. **Overall, CoPiC can adapt a broad range of temperature settings (0.2 - 0.8), thereby ensuring both the diversity and quality of the plans.**
>
>     | Method | Pick | Examine | Clean | Heat | Cool | Pick Two |
>     |---------|---------|---------| --------- | --------- | --------- | --------- |
>     | CoPiC (temperature=0.2) | 100.00 (0.12M) | 90.00 (0.08M) | 92.90 (0.63M) | 97.39 (0.57M) | 95.24 (0.38M) | 92.94 (0.11M) |
>     | CoPiC (temperature=0.5) | 100.00 (0.13M) | 95.56 (0.07M) | 91.61 (0.55M) | 96.52 (0.48M) | 100.00 (0.37M) | 96.47 (0.09M) |
>     | CoPiC (temperature=0.6) | 100.00 (0.05M) | 100.00 (0.04M) | 100.00 (0.33M) | 100.00 (0.26M) | 100.00 (0.28M) | 95.29 (0.06M) |
>     | CoPiC (temperature=0.8) | 100.00 (0.10M) | 100.00 (0.07M) | 99.35 (0.39M) | 94.78 (0.52M) | 91.43 (0.29M) | 97.65 (0.07M) |
>
> - **Quality**. CoPiC is capable of iteratively constructing high-quality plans. **Although the quality of the plans is influenced by the specific base LLM used, CoPiC can adapt to different LLMs, as shown in Table 3 in the paper**. This adaptability is due to CoPiC's construction of interaction contexts based on the domain-adaptive selector, which instructs the LLM to evolve planning programs and thereby incrementally improves the quality of the plans.
>
> - **The Number of Generated Plans $ n $**. A small $ n $ leads to an overall lower quality of candidate plans, while an excessively large $ n $ results in an overly expansive planning space for the domain-adaptive selector, which affects its training. Therefore, **both too small and too large values of $ n $ can reduce the success rate and increase costs**. According to the ablation study on $ n $ presented in Appendix Table 9, $ n = 3 $ is the optimal value.

---

### Official Review · Reviewer_rsyp · 2025-11-01

**Soundness:** 3
**Presentation:** 3
**Contribution:** 3
**Rating:** 4
**Confidence:** 5

**Summary:**

This paper introduces CoPiC, a query-efficient LLM planner that generates various code plans before having the best plan selected by a domain-active selector, executed in the environment, and replan until success. The domain-active selector is finetuned via PPO on each domain’s training sets before inference. Results are presented on ALFWorld, NetHack, and StarCraft II Unit Building showing that CoPiC achieves superior performance at ultra-low cost.

**Strengths:**

$Originality$: This paper essentially introduces a method that combines code planning and a BoN judge that selects the best code plan at each step; however the judge is fine-tuned on the domain which addresses past issues with generalization to new domains.

$Quality$: The performance of the method is very high compared to past methods at very low cost.

$Clarity$: The motivation of the paper is clear and the method and results are clear to understand.

$Significance$: LLM planning is very query-inefficient due to dense interactions at each step making bar to entry difficult; this method introduces an alternative code planning approach that brings cost down while also boosting performance.

**Weaknesses:**

- This approach requires a lot of setup to be applied to a new domain compared to the zero-shot approaches tested on in the paper including
  - Finetuning to a training dataset which may not be available; experiment 5.4.3 demonstrates the necessity of the selector making this a requirement
  - Prompt engineering the structure of the code to ensure the agent can plan via code which requires domain expertise; the prompts in Appendix H include many examples for code including
    - Assistant methods for ALFWorld in H.1.1
    - Heavy prompt engineering for Nethack and StarCraft 2 in H.2.1 and H.3.2 including data imports and code structure whose structure shows in I.2 and I.3 for code evolution

**Questions:**

My main concern is that the high performance is due mainly to the effort put into initial prompts given that the code evolutions are heavily biased towards the code structure in the prompt.
  - The ablation in 5.4.3 seems to convey that the selector is a necessary part for CoPiC to work; however, the baseline without the selector seems to fail at every other trial. This looks like a code formatting bug, why is this happening?
  - Have the authors considered generating full-length trajectories instead of code plans? This maintains cost-effectiveness while improving ease of adoption to new domains

---

> ### Author Response · Authors · 2025-11-20
> **Rebuttal -- Part 1**
>
> Thanks for your detailed review. We are glad to discuss your concerns one by one. Any further discussion will be appreciated.
> Besides, we have revised our paper to address your concerns. We summarize the main changes as the following:
>
> (1) Add the result of CoPiC(TSL)—that is, CoPiC with Test Set Learning—in Table 1. (**Weakness**)
>
>
> > **Weakness**. This approach requires a lot of setup to be applied to a new domain compared to the zero-shot approaches tested on in the paper including:
> (1) Finetuning ...
> (2) Prompt engineering ...
> **Question 1**. The high performance is due mainly to the effort put into initial prompts.
>
>
> CoPiC can be easily applied to new domains. And compared to the baselines, CoPiC exhibits superior generalization capabilities. We address your question from two perspectives: training and prompt.
>
> - **Training.** We emphasize that **the baselines are not zero-shot; instead, they perform online learning on the test set**. In contrast, CoPiC learns on the training set and then **generalizes to the test set**. Therefore, **CoPiC achieves superior performance under a more stringent setup**. Moreover, CoPiC can also conduct online learning on the test set, similar to the baselines, and **does not necessarily require an additional training set**. Below are the results of CoPiC performing online learning on the test set, with table entries formatted as **Success Rate (Cost)**. It can be observed that under an identical experimental setup, CoPiC still achieves a higher success rate than the baselines by 17.17% and reduces token consumption by 87.16%.
>
>     | Method | Pick | Examine | Clean | Heat | Cool | Pick Two |
>     |---------|---------|---------| --------- | --------- | --------- | --------- |
>     | CoPiC | 100.00 (0.04M) | 100.00 (0.04M) | 100.00 (0.29M) | 96.52 (0.27M) | 100.00 (0.25M) | 100.00 (0.09M) |
>     | AdaPlanner | 100.00 (0.63M) | 64.44 (1.95M) | 91.61 (0.89M) | 76.52 (0.74M) | 89.52 (1.57M) | 87.06 (1.58M) |
>     | Reflexion | 91.67 (2.09M) | 86.67 (1.51M) | 73.55 (2.54M) | 75.65 (1.88M) | 73.33 (1.46M) | 81.18 (1.65M) |
>     | Prospector | 70.83 (1.74M) | 65.56 (1.18M) | 75.48 (2.15M) | 82.61 (1.48M) | 81.90 (1.18M) | 71.76 (1.30M) |
>     | REPL-plan | 82.50 (0.66M) | 83.33 (0.41M) | 89.03 (0.75M) | 91.30 (0.61M) | 88.57 (0.47M) | 100.00 (0.43M) |
>
> - **Prompt.** CoPiC's high performance is not attributable to the effort put into the initial prompts, but rather to the core design of CoPiC, namely **the iterative refinement of planning programs and domain-adaptive selector**. The evidences are as follows:
>     - Our program examples **do not require domain expertise**; they merely define **the input and output of the programs**, which is **the essential information for LLMs to generate programs**. For instance, in SC2, we only define the structure of the observation information as the input to the program and specify that the output of the program is a sequence of actions available in the environment.
>     -  The results in Figure 3 and Figure 4(a) both demonstrate that **without the improvement of planning programs, relying solely on initial prompts is insufficient to complete the tasks**, achieving only an initial performance similar to that of the baselines (AdaPlanner, Reflexion).
>
>     - The results in Figure 4(b) demonstrate that **without the optimization of the domain-adaptive selector, relying solely on the evolution of the programs yields a less stable result**, thereby degrading the performance of CoPiC.
>     - Using programs as examples to construct prompts is **commonly used**. **The prompt structure of CoPiC is identical to that of AdaPlanner in the baselines**. Both the structure of the prompt used in CoPiC and AdaPlanner consist of: a system prompt (some background information, such as "You are a household agent" or "You are an agent who plays the rogue-like game NetHack"), examples (programs serving a few-shot purpose), and a question (the current task). This further illustrates that **the high performance of CoPiC is derived from its design, rather than from the initial prompts**.

---

> ### Author Response · Authors · 2025-11-20
> **Rebuttal -- Part 2**
>
> > Question 2. The ablation in 5.4.3 seems to convey that the selector is a necessary part for CoPiC to work; however, the baseline without the selector seems to fail at every other trial. This looks like a code formatting bug, why is this happening?
>
>
> - Figure 4(b) in Section 5.4.3 indicates that the absence of a selector renders CoPiC **less stable**. However, this **does not lead to complete failure** but rather **increases randomness and degrades performance**.
>
> - The failures of CoPiC in the absence of a selector **are not due to code formatting bugs but rather stem from incorrect plan choices and inefficient evolution of planning programs**, both of which result from the lack of a domain-adaptive selector. The domain-adaptive selector is necessary, as it aligns the selection of the plan with long-term rewards. Without the domain-adaptive selector, the randomly chosen plans are not the most suitable among the candidates, thereby affecting task completion. Moreover, the interaction context constructed based on these suboptimal plans reduces the efficiency with which the LLM can correct planning programs, further impairing CoPiC's performance.
>
>
> > Question 3. Have the authors considered generating full-length trajectories instead of code plans? This maintains cost-effectiveness while improving ease of adoption to new domains
>
> - We clarify that the cost of generating a planning program is **comparable** to that of generating a full-length plan. The planning program iteratively generates plans based on observation **without the need to query the LLM again, thereby incurring no additional overhead**.
>
> - Moreover, **compared with full-length plans, planning programs exhibit greater adaptability**. Due to the partially observable nature of the environment and the LLM's lack of knowledge about the environment, it is challenging for the LLM to generate static, full-length plans in a single query to complete tasks. In contrast, planning programs can dynamically generate plans based on the current observations from the environment, thereby offering stronger adaptability.

---

> ### Author Response · Authors · 2025-12-01
> **Rebuttal -- Part 3**
>
> > The high performance is due mainly to the effort put into initial prompts given that the code evolutions are heavily biased towards the code structure in the prompt.
>
> We emphasize that CoPiC's high performance **should not be attributable to the effort put into initial prompts**. Two main evidence support this claim.
>
> - **CoPiC imposes weaker demands on the prompt than the baselines.** The program example put in CoPiC's prompt are drawn from a task category that **differs from** the target task (e.g., a Pick example is provided when the agent faces a Clean task). By contrast, the baselines' prompt (including those of AdaPlanner and Reflexion) contain examples that belong to the **same** task category as the target (e.g., Clean lettuce is shown when the agent faces Clean apple). Consequently, CoPiC works under a stricter 'zero-shot' condition with respect to task type, demonstrating that its gains stem from the **core mechanism**, i.e., iterative refinement of planning programs and domain-adaptive selector, rather than from prompt design.
>
> - **CoPiC still outperforms baselines under a 'minimal-example' without much expert effort.** To directly address the reviewers' concern, we conducted an additional 'minimal-example' experiment on StarCraft II Unit Building (SC2). Specifically, we replaced the planning program example in CoPiC's prompt with the simplest possible SC2 task -- Building SCV (i.e. **the example provided in Sec 4.1.1**) -- thereby minimizing the environmental information conveyed by the prompt. The results are summarized in the table below, with table entries formatted as **Success Rate (Cost)**. As can be seen, (1) CoPiC(minimal-example) is almost **identical** to CoPiC in both success rate (97.33% vs. 98.67%) and cost (0.90M vs. 0.88M); (2) CoPiC(minimal-example) still achieves a **45.17%** higher success rate and **64.70%** lower cost than the baselines. This outcome confirms that CoPiC's high performance is rooted in its architectural innovations, not in prompt engineering. Besides, we have added this 'minimal-example' experiment in **Appendix E.13**.
>
>     | Method | Medium | Hard | SuperHard |
>     |-------|-------|-------| ------- |
>     | CoPiC (minimal-example) | 100.00 (0.18 M) | 100.00 (0.08M) | 92.00 (2.45M) |
>     | CoPiC | 100.00 (0.17 M) | 100.00 (0.06M) | 96.00 (2.42M) |
>     | AdaPlanner | 64.00 (0.53M) | 71.00 (0.36M) | 32.00 (6.08M) |
>     | Reflexion | 64.00 (0.98M) | 24.00 (0.78M) | 34.00 (6.74M) |
>     | Prospector | 71.00 (0.95M) | 68.00 (0.68M) | 42.00 (7.02M) |
>     | REPL-plan | 59.00 (0.59M) | 58.00 (0.39M) | 39.00 (5.61M) |

---

### Official Review · Reviewer_9w8c · 2025-11-01

**Soundness:** 3
**Presentation:** 4
**Contribution:** 3
**Rating:** 6
**Confidence:** 3

**Summary:**

The paper proposes CoPiC (Code-Driven Planning with Domain-Adaptive Selector): it uses multiple LLM-generated “planning programs” (MoE-style experts) as a low-cost plan supply, paired with a domain-adaptive selector that, at each step given the environment observation, scores candidate plans and picks the best. It then leverages interaction history to “evolve” the programs, and fine-tunes the selector with RL using LoRA+PPO, forming a closed loop of “generate–select–learn.” Compared with high-interaction step-by-step querying, CoPiC balances long-horizon return alignment with lower query cost, achieving on average +19.14% success rate and −79.39% token cost on ALFWorld, NetHack, and StarCraft II Unit Building. Ablations removing the selector / program evolution / varying the number of experts validate the contribution of each module. The core contribution is integrating code-based planning programs with a learnable selector, providing (i) a reusable, evolvable plan representation, (ii) a selection mechanism oriented to long-term returns, and (iii) a low-cost path to cross-environment generalization.

**Strengths:**

Originality: Connects “multi planning programs (MoE-like experts) → domain-adaptive selector → program evolution + selector RL fine-tuning” into a closed-loop paradigm, unifying generate–select–learn; using executable programs to carry plans enables continual evolution and reuse, distinct from single-trajectory progression or heuristic voting.

Quality: Delivers significant success-rate improvements alongside large token-cost reductions across text-world, roguelike, and strategy-building environments; includes key ablations (selector removal, expert-count sweep, evolution curves) and multi-seed statistics. The pipeline, pseudocode, prompt templates, and hyperparameters are public, forming a solid evidence chain.

Clarity: Top-down exposition (motivation → framework → training/inference → experiments/ablations). Core concepts are defined at first mention; the overview figure clearly depicts data flow; main tables use unified metrics (SR/Step/Cost); evolution/ablation curves localize sources of gain and cost components, facilitating reproduction.

Significance: Provides a general and practical solution path to “high interaction cost, short-sighted feedback, domain gap”; effective on both closed- and open-source bases, nudging the field from “frequent-dialogue agents” toward “low-interaction, strong planning, evolvable” paradigms.

**Weaknesses:**

There is a setup mismatch on ALFWorld—CoPiC learns on the training set and is evaluated zero-shot on the test set, whereas some baselines (e.g., Reflexion/AdaPlanner) perform online learning/adaptation on the test set. The paper argues fairness by claiming a similar number of learned tasks, but differences in adaptation timing and data distribution introduce confounders that compromise comparability. Since a core conclusion is “under fair conditions, we improve success rate while significantly cutting cost,” inconsistent training/evaluation paradigms—even with a similar number of tasks—can over- or under-estimate any method. I suggest adding a replication with fully aligned protocols: (i) unified training/evaluation paradigms (“train-only learning” vs. “test-time online learning”), (ii) unified base models/variants, (iii) shared seeds and hardware/concurrency, to remove bias from protocol differences.

**Questions:**

Possible insufficiency of candidate “diversity”: The paper treats n>1 as evidence of diverse candidates but does not explain how to prevent highly similar/duplicate plans across experts (e.g., decoding-parameter differences, template preferences, anti-duplication constraints, similarity filtering/reranking). Diversity metrics and controls are also missing.

Unclear handling of failures and exceptions: The algorithm appears to replan stepwise (“regenerate and reselect each step”), but it’s unspecified whether mid-execution failures or unmet preconditions trigger local repairs versus whole-plan replacement, and what the retry budget and triggers are.

---

> ### Author Response · Authors · 2025-11-20
> **Rebuttal -- Part 1**
>
> Thanks for your detailed review. We are glad to discuss your concerns one by one. Any further discussion will be appreciated.
> Besides, we have revised our paper to address your concerns. We summarize the main changes as the following:
>
> (1) Add the result of CoPiC(TSL)—that is, CoPiC with Test Set Learning—in Table 1. (**Weakness**)
>
> (2) Add the Quantification of Diversity of planning programs in Appendix E.10. (**Question1**)
>
>
> > Weakness. Setup mismatch of CoPiC and baselines.
>
> - We emphasize that, compared to the experimental setup of the baselines, CoPiC's setup imposes higher demands on the model, **requiring it to generalize zero-shot to unseen tasks**. Therefore, **CoPiC outperforms the baselines under more stringent setup**.
>
>     - The paradigm of CoPiC (planning programs + domain adaptive selector) enables it to learn on the training set, constructing "planning programs + selector" for each category of tasks, and then **generalizing zero-shot** to the same category of tasks in the test set. In contrast, the paradigm of the baselines (planning for specific tasks) **restricts them to learning only on the test set**.
>
>     - CoPiC and the baselines utilized **the same base models**, including GPT-3.5 (Table 1), GPT-4o (Table 2), DeepSeek-Coder, DeepSeek-V3, and Qwen2.5-14B (Table 3).
>
>     - CoPiC and the baselines employed **the same seeds and hardware/concurrency** settings.
>
> - To further address your concern, we constructed an experimental setup **identical to that of the baselines** by allowing CoPiC to perform online learning on the test set. The results are presented below, with table entries formatted as **Success Rate (Cost)**. Under this completely consistent setup, CoPiC still achieved a success rate that was 17.17% higher than the baselines, while reducing token consumption by 87.16%.
>     | Method | Pick | Examine | Clean | Heat | Cool | Pick Two |
>     |---------|---------|---------| --------- | --------- | --------- | --------- |
>     | CoPiC | 100.00 (0.04M) | 100.00 (0.04M) | 100.00 (0.29M) | 96.52 (0.27M) | 100.00 (0.25M) | 100.00 (0.09M) |
>     | AdaPlanner | 100.00 (0.63M) | 64.44 (1.95M) | 91.61 (0.89M) | 76.52 (0.74M) | 89.52 (1.57M) | 87.06 (1.58M) |
>     | Reflexion | 91.67 (2.09M) | 86.67 (1.51M) | 73.55 (2.54M) | 75.65 (1.88M) | 73.33 (1.46M) | 81.18 (1.65M) |
>     | Prospector | 70.83 (1.74M) | 65.56 (1.18M) | 75.48 (2.15M) | 82.61 (1.48M) | 81.90 (1.18M) | 71.76 (1.30M) |
>     | REPL-plan | 82.50 (0.66M) | 83.33 (0.41M) | 89.03 (0.75M) | 91.30 (0.61M) | 88.57 (0.47M) | 100.00 (0.43M) |
>
> > Question2. Unclear handling of failures and exceptions. What the retry budget and triggers are.
>
> - We categorize failures and exceptions into three types for discussion:
>
>     - **Errors in Programs**: These include syntax errors in the program, access to undefined variables, etc. For such errors, CoPiC re-queries the LLM to regenerate planning programs and replaces the erroneous programs. The token cost incurred by this re-querying **has been accounted** for in the reported results in the paper.
>
>     - **Invalid Actions Returned by Planning Programs**: In this case, CoPiC replaces invalid actions with no-ops during the actual interaction with the environment.
>
>     - **Planning Failures to Complete Tasks**: Instead of performing local repairs, CoPiC conducts **whole-plan replacement** by resetting the task and regenerating the planning programs, rather than regenerating planning programs from an intermediate state of the task.
>
>
> - **Retry Budget and Triggers:** The mechanism for the retry budget and triggers is detailed in Algorithm 1 (Appendix G). Specifically, a set of generated planning programs is applied to accomplish N tasks. If the success rate falls below a predefined threshold (line 24 of the Algorithm 1), the evolution of planning programs is **triggered**. Additionally, the number of such evolution is constrained by a maximum limit, denoted as max_Trials (line 28 of the Algorithm 1), which defines the **retry budget**. The values of N, the threshold, and max_Trials are provided below and have also been specified in Table 5 of Appendix D.2 in the paper.
>
>
>     | Hyperparameters | N | threshold | max_Trials |
>     |---------|---------|---------| --------- |
>     | Value | 20 | 0.9 | 10 |

---

> ### Author Response · Authors · 2025-11-20
> **Rebuttal -- Part 2**
>
> > Question1. Possible insufficiency of candidate "diversity". Diversity metrics and controls are also missing.
>
> When generating planning programs, we set a relatively high temperature (0.6) for the LLMs and employed multiple sampling to produce these programs. **Given the large planning space of the tasks and the lack of domain-specific experience in general-purpose LLMs**, the programs generated through multiple sampling exhibit sufficient diversity. The table below presents the quantitative diversity results for a set of planning programs generated by each base model on the Pick and Examine tasks in ALFWorld. The calculation formula is:
> $$
> \text{Diversity} = 1 - \frac{2}{N(N-1)} \sum_{i=1}^{N-1} \sum_{j=i+1}^{N} \text{sim}(\rho_i, \rho_j)
> $$
> where $\text{sim}(\rho_i, \rho_j)$ denotes the MinHash Jaccard Similarity between two planning programs. The results in the table, with each entry formatted as **Mean Diversity (Max Diversity)** for three sets of planning programs, each set comprising three programs generated by the LLM for the task, demonstrate that **the planning programs generated by CoPiC exhibit good diversity**.
>
> | Tasks | GPT-3.5 | DeepSeek-Coder | DeepSeek-V3 | Qwen2.5-Coder-14B-Instruct | Qwen2.5-14B-Instruct|
> |---------|---------|---------| --------- | --------- | --------- |
> | Pick | 0.2153 (0.2682) | 0.3516 (0.4167) | 0.3455 (0.3750) | 0.4193 (0.5208) | 0.2023 (0.3672) |
> | Examine | 0.3342 (0.4062) | 0.3177(0.4453) | 0.3681 (0.3776) | 0.3741 (0.3229) | 0.3741 (0.4453) |
>
> For the same task, planning programs generated by the LLM may share the same overall logic but differ in specific implementations of particular steps. For example, in the Cool task, both programs follow the general logic of "find object $\rightarrow$ pick object $\rightarrow$ cool object $\rightarrow$ find receptacle $\rightarrow$ place object", yet **they exhibit difference in the specific implementation of a particular step**, such as "finding object". For example, below are two concrete planning programs, with **Diversity = 0.2813. The first program conducts simple exploration when the object is not found, while the second attempts to traverse potential receptacles first, only resorting to exploration if no suitable receptacles are available**. Such differences arise due to the complexity of the tasks and contribute to the diversity among planning programs.
> ```python
> def pick_cool_then_place(self, objecttype, receptacletype):
>     # Step 1: Find and Pick the object
>     if self.name2type(self.holding) != self.name2type(objecttype):
>         # Find the object
>         entity = self.find_object(objecttype)
>         if entity is None:
>             # Explore the environment to find the object
>             self.explore()
>             return
>         else:
>             # Go to the object's receptacle and take it
>             self.goto(entity.in_on)
>             r_entity = self.seen_entitys[entity.in_on]
>             if r_entity.openable and not r_entity.isopen:
>                 self.open_receptacle(r_entity.name)
>             self.take(entity.name, entity.in_on)
>             return
>     else:
>         # Step 2: Cool the object in a fridge
>         ...
>         # Step 3: Find specified receptacle and Place the cooled object in it
>         ...
> ```
> ---
> ```python
> def pick_cool_then_place(self, objecttype, receptacletype):
>     # Step 1: Find and Pick the object
>     if self.name2type(self.holding) != self.name2type(objecttype):
>         # Find the object
>         entity = self.find_object(objecttype)
>         if entity is None:
>             # Find receptacletypes that can contain the object
>             r_types = self.find_canbe_contained(objecttype)
>             if r_types is None:
>                 # Explore the environment to find the object
>                 self.explore()
>                 return
>             else:
>                 # Check each receptacletype to find the object
>                 for r_type in r_types:
>                     r_entitys = self.find_receptacles(r_type)
>                     if r_entitys is None:
>                         continue
>                     for r_entity in r_entitys:
>                         self.goto(r_entity.name)
>                         if r_entity.openable and not r_entity.isopen:
>                             self.open_receptacle(r_entity.name)
>                         # Check if the object is inside the receptacle
>                         if entity is None:
>                             entity = self.find_object(objecttype)
>                         if entity is not None:
>                             break
>                     if entity is not None:
>                         break
>                 if entity is None:
>                     # Explore the environment to find the object
>                     self.explore()
>                     return
>         # Go to the object's receptacle and take it
>         ...
>     else:
>         # Step 2: ...
>         ...
>         # Step 3: ...
>         ...
> ```

---

> > ### Comment · Reviewer_9w8c · 2025-11-26
> >
> > Thanks for your thoughtful response. I will keep my score

---

> > > ### Author Response · Authors · 2025-11-26
> > >
> > > Thank you again for your thoughtful review. We would like to know if our response has addressed your concerns. If so, could you please update the score? We welcome further discussion.

---

### Author Response · Authors · 2025-11-20
**Global Response**

We sincerely thank all reviewers for their time and valuable feedback. We are encouraged that the reviewers found our work to have an excellent presentation (Reviewer 9w8c), clear motivation (Reviewer rsyp), convincing results (Reviewer 9w8c, rsyp, 4nz9, R9TH), as well as a general and practical core contribution, namely the iterative optimization of planning programs and the domain-adaptive selector (Reviewer 9w8c).

### **Summary of Revision**
In response to the questions raised by each reviewer, we have revised our paper. The summary of these revisions is as follows.

- For Reviewer 9w8c
    - Add the result of CoPiC(TSL)—that is, CoPiC with Test Set Learning—in **Table 1** to illustrate that CoPiC still outperforms baselines under the same experimental setup.
    - Add the Quantification of Diversity of planning programs in **Appendix E.10** to illustrate that the planning programs generated by CoPiC are sufficiently diverse.
- For Reviewer rsyp
    - Add the result of CoPiC(TSL)—that is, CoPiC with Test Set Learning—in **Table 1** to illustrate that CoPiC can also conduct online learning on the test set, similar to the baselines.
    - Add an additional **'minimal-example'** experiment on StarCraft II Unit Building in **Appendix E.13**, which shows that CoPiC maintains its performance advantage over baselines with minimal expert effort.
- For Reviewer 4nz9:
    - Add a new baseline, the PDDL-based method LLM-DP, in **Table 1**.
    - Add an analysis of the impact of temperature variations on CoPiC's performance in **Appendix E.8**.
- For Reviewer R9TH:
    - Add the result of CoPiC(LaJ)—that is, CoPiC taking LLM-as-a-Judge as the selector—in **Table 1** to illustrate the necessary of domain-adaptive selector.
    - Add a new baseline, TWOSOME, in **Table 1**.
    - Add a simple, complete, and intuitive example of planning program in **Section 4.1.1** for aiding understanding.
    - Add an analysis of the impact of experiences from other type of tasks on CoPiC's initial performance in **Appendix E.9**.
    - Add an analysis of how the number of planning programs affects performance on the SuperHard task in **Appendix E.11**.
    - Add an analysis of the selector’s frequency in switching planning programs in **Appendix E.12**.


We believe these revisions wil significantly strengthen the paper and directly address the concerns. Below, we responsed to each reviewer's points in detail.

---

### Meta-Review · Area_Chair_GSTX · 2026-01-13

**Summary:**

This paper proposes a query-efficient LLM planning framework that generates multiple executable “planning programs,” selects among their candidate plans via a domain-adaptive selector trained with RL, and iteratively evolves the programs using interaction experience. Across ALFWorld, NetHack, and StarCraft II Unit Building, reviewers agreed the approach is well-motivated and generally strong, delivering substantial success-rate gains while drastically reducing token cost relative to interaction-heavy baselines. Reviewer feedback after rebuttal indicates concerns were largely resolved or mitigated, ACs recommend accept.

**Reviewer Concerns:**

There are concerns about evaluation comparability, prompt dependence, missing baselines, and insufficient ablations/analysis. Reviewer feedback after rebuttal indicates concerns were largely resolved or mitigated.

**Reviewer Scores:**

During rebuttal, the authors addressed the key reviewer concerns with targeted additional experiments and clarifications, and they revised the paper accordingly. ACs recommend accept.

---

### Decision · Program_Chairs · 2026-01-26

Accept (Poster)